# Future Economic Perspective and Potential Revenue of Non-Subsidized Wind Turbines in Germany

Lucas Blickwedel[1], Freia Harzendorf[1], Ralf Schelenz[1] and Georg Jacobs[1]

[1] Chair for Wind Power Drives, RWTH-Aachen University, Aachen, Germany

*Correspondence to*: Lucas Blickwedel (lucas.blickwedel@cwd.rwth-aachen.de)

**Abstract.** Fixed feed-in tariffs based on the Renewable Energy Act grant secure revenues from selling electricity for wind turbine operators in Germany. Anyhow, the level of federal financial support is being reduced consecutively. Plant operators must trade self-sufficiently in the future, hence generate revenue by selling electricity directly on electricity markets. Therefore, uncertain future market price developments will influence investment considerations and may lead to stagnation in the

expansion of renewable energies. This study estimates future revenue potentials of non-subsidized wind turbines in Germany to reduce this risk. The paper introduces and analyses a forecasting model that generates electricity price time series suited for revenue estimation of wind turbines based on the electricity exchange market. Revenues from the capacity market are neglected. The model is based on openly accessible data and applies a merit-order approach in combination with a simple agent-based approach to forecast long-term day-ahead prices at an hourly resolution. The hourly generation profile of wind

turbines can be mapped over several years in conjunction with fluctuations in the electricity price. Levelized Revenue of Energy are used to assess both dynamic variables (electricity supply and price). The merit order effect from the expansion of renewables as well as the phasing out of nuclear energy and coal are assessed in a scenario analysis. Based on the assumptions made, the opposing effects could result in a constant average price level for Germany over the next twenty years. The influence of emission prices is considered in a sensitivity analysis and correlates with the share of fossil generation capacities in the

generation mix. In a brief case study, it was observed that current average wind turbines are not able to yield financial profit over their lifetime without additional subsidies for the given case. This underlines a need for technical development and new business models like Power Purchase Agreements. The model results can be used for setting and negotiating appropriate terms, such as energy price schedule or penalties for those agreements.

## 1 Introduction

Renewable electricity generation has increased exponentially in Germany over the last few decades due in large part to subsidization by the Renewable Energy Act (Erneuerbare Energien Gesetz, EEG). The law was enacted to achieve the policy goal of reducing the energy sectors greenhouse gas emissions. Fixed feed-in tariffs secured revenues from selling electricity for renewable energy sources and ensured independence from the fluctuating electricity exchange price. Thus, investments in the renewable energy sector were particularly attractive and led to a strong expansion. By introducing a tendering procedure,

an attempt was then made to increase competition and promote the competitiveness of wind energy. In the future however, subsidies will probably cease completely. Electricity from renewable energy sources will have to be sold in alternative ways to generate revenue. These are likely to be based on the electricity exchange markets. Hence, the electricity exchange price at the time of electricity generation will become a crucial factor for profitability next to the already widely considered cost of electricity generation (Federal Ministry for Economic Affairs and Energy, 2019a, 2019b, 2019c).

The aim of this research is to address and quantify the barrier of uncertain revenues of electricity generation from wind turbines (WT) without subsidies in Germany. How can future electricity exchange prices be estimated and put in perspective to the generated electricity? To answer this question, the markets of Germany and its neighboring countries are considered within the framework of a new forecasting model, which will be developed and discussed in this paper.

First, a literature research for three different aspects relevant to this task is conducted: In Section 1.1 the current market
situation and potential revenue sources for WT operators in Germany are assessed with a focus on long-term Power Purchase Agreements as one possible alternative for selling electricity. Afterwards, in Section 1.2 different metrices to evaluate the economic efficiency and perspective of renewable energy sources are being discussed. Section 1.3 will then address existing approaches with a similar scope for modelling and forecasting electricity supply systems and exchange prices.

Based on the literature research, the elaborated model requirements and forecasting model will be derived and discussed in
Section 2. Finally, in Section 3 model results are presented and interpreted along a brief case study and conclusion, followed by a critical discussion of the model and outlook in Section 4.

**1.1 Potential revenue sources for wind turbine operators in Germany**

Since 2017 the level of subsidization for electricity from WTs in Germany is determined through tendering with a pay-as-bid system, regulated by the German Renewable Energy Act. After successfully taking part in the tendering procedure, WT
operators receive individual feed-in tariffs per kWh according their bid. The Renewable Energy Act defines a maximum tender volume for each year. Bids that exceed the set limit are not receiving financial support (Deutscher Bundestag, 2020). There are three main options for plant operators in Germany who do not receive subsidies to sell their generated electricity: Selling directly on the electricity exchange market, through bilateral contracts or by providing generation capacity on the capacity market. The latter naturally does not pose a significant source of revenue for the variable electricity generation from wind and
solar energy so far and will therefore be disregarded for this paper. In case of selling directly on the electricity exchange market, revenues depend largely on the temporal development of the exchange price and local wind conditions. The question of how their course over a longer period of time can be estimated is the core element of this research and the following chapters. An alternative for selling electricity from renewable sources on the electricity exchange markets are so called Power Purchase Agreements (PPA). PPAs are mostly made between corporate electricity consumers and plant operators. They enable bilateral
trading including consultation between contracting parties. Those agreements normally cover a period of up to ten years and are established individually each time by the contracting parties. PPAs define the following aspects of power purchase: amount of electricity, price, contract terms and penalties for breach of contract (Javadi et al., 2011; Elwakil and Hegab, 2018 - 2018).

PPAs already have a significant market share in the sale of electricity from renewable energy sources in some countries around the world. In particular in the US, the market for PPAs has been growing in recent years. Energy intensive companies such as Google, Microsoft and Facebook have committed to 100% green electricity. Google has already concluded PPAs for a total of about 1.8 GW. In Brazil, energy-intensive consumers have been able to conclude PPAs for electricity from conventional or renewable energy sources since 1995 (Coussi and Harada, 2020; Berger et al., 2016). Compared to other European countries the PPA market in Germany is much less developed. In a lot of neighboring countries PPAs are already an established procedure (Fischer et al., 2019; Tang and Zhang, 2019). Sweden (33%), Norway (30%) and the UK (16%) are the European countries with the highest share of PPAs (Klinger 2019). However, the majority of renewable energies in Europe, including Germany, were supported through a fixed feed-in tariff financed by public funds (Berger et al., 2016). Since many plants will be excluded from subsidies from 2020 onwards, PPAs are increasingly being considered in Germany.

Although PPAs can be negotiated independently of the electricity exchange price, it is nevertheless common for the contracting parties to base their agreement on the development of the electricity exchange price. PPAs are therefore considered as an alternative selling option in this paper. Consequently, the two potential revenue sources to be considered in this paper are direct sales through the electricity exchange market and bilateral sales through PPAs. The future electricity exchange price seems to be the correct forecast object to derive statements about both potential revenue sources.

### 1.2 Existing metrics for evaluating the economic value of wind turbines

Several different metrics are known by literature to describe the economic efficiency of electricity generation techniques and plants. Generally, the economic efficiency of a WT can be assessed based on three essential quantities: Cost, revenues and electricity yield. A distinction can be made between those metrics that consider the costs of a generation technology and those that consider the revenue situation. Those metrics that are found to be relevant for evaluating potential revenues within this study are listed in Table 1 and further described in the following.

The overall performance of WT (or other generation plants) is often evaluated based on the levelized cost of energy (LCOE). The LCOE are defined as the total lifetime cost over the total lifetime energy production. It is a widely used and simple metric for estimating the value of renewable energy sources projects (Campbell et al., 2009 - 2009; Boubault et al., 2016; Ouyang and Lin, 2014; Parrado et al., 2016). To calculate the LCOE, information on annual costs and annual electricity generation is needed. The annual costs consider investment, O&M and fuel costs. However, the LCOE do not consider the time-varying value of electricity generation, which is already criticized in literature (Hirth and Steckel, 2016; Hirth, 2013; Simpson et al., 2020). Therefore, the LCOE are no holistic indicator for the profitability and economic value of a plant and its potential revenue. Its value can only be interpreted as the minimum fixed revenue required for an economical plant operation.

**Table 1: Overview of relevant metrices for evaluating the economic value of electricity generation techniques**

| Metric | Abbreviation | Objective | Formula |
|---|---|---|---|
| Levelized Cost of Energy | LCOE | Cost | $\sum_{t=1}^{n} \dfrac{C_t}{(1+i)^t} \Big/ \sum_{t=1}^{n} \dfrac{W_{t,el}}{(1+i)^t}$ |
| System Levelized Cost of Energy | sLCOE | Cost | $\sum_{t=1}^{n} \dfrac{C_t}{(1+i)^t} \Big/ \sum_{t=1}^{n} \dfrac{W_{t,el}}{(1+i)^t} + \dfrac{d}{dW_{t,el}} C_{int}$ |
| Levelized Revenue of Energy | LROE | Revenue | $\sum_{t=1}^{n} \dfrac{R_t}{(1+i)^t} \Big/ \sum_{t=1}^{n} \dfrac{W_{t,el}}{(1+i)^t}$ |
| Levelized Avoided Cost of Energy | LACE | Revenue | $\sum_{j=1}^{Y} R_{j,el} * h_j + R_{j,cap} \Big/ \sum_{j=1}^{Y} h_j$ |
| Simplified Levelized Avoided Cost of Energy | LACEs | Revenue | $\sum_{j=1}^{Y} R_{j,el} * h_j \Big/ \sum_{j=1}^{Y} h_j$ |

| | | | | | | | |
|---|---|---|---|---|---|---|---|
| $C$ | Cost | $W_{el}$ | Electricity generation | $R_{el}$ | Revenue from electricity sales | $h$ | Dispatched hours |
| $C_{int}$ | Integration cost | $i$ | Interest rate | $R_{cap}$ | Revenue from capacity market | $Y$ | Time periods |

The System LCOE (sLCOE) metric addresses this problem by adding marginal integration costs. Uckerdt et al. define a method to determine the marginal integration costs from annual electricity demand, generation from renewable sources and system

costs with and without renewables (Ueckerdt et al., 2013). This makes the metric interesting for a systemic view, but is not suitable for the centralized planner, since revenues are not considered and complex simulation of integration costs is required (Joskow, 2011; Lucheroni and Mari, 2018; Reichenberg et al., 2018).

Analogous to the LCOE, the LROE measure revenues instead of costs of a plant. By combining LROE and LCOE the net present value (NPV = LROE - LCOE) can be calculated. Thereby, information is provided on how incentives have to be set to

encourage growth of renewable energies. However, timeseries data and revenues from electricity & capacity sales as well as incentives must be considered when calculating revenues (Baker, 2011b, 2011a). The LROE can be regarded as the expectable average revenue in a given market. LROE vary for the same plant in different markets. The additional market information poses an advantage of LROE over LCOE. Unlike LROE, LACE does not consider revenues from incentives, but from the capacity market. This makes modeling a bit less complicated and it is more interesting for centralized planners than from a

systemic point of view. Due to the neglection of the capacity market in this paper, the LACE appears to be less suitable for the subject under consideration. The LACEs further simplify revenues by incorporating capacity sales, which generally represent only a small part of RES revenues. Revenues are calculated on the assumption of a linear relationship between spot market price and residual demands. This simplifies the simulation even further, while still taking into account that the value of electricity generation varies over time (Simpson et al., 2020).

In order to evaluate the potential revenue from wind energy, a measurand must be chosen that considers revenues instead of costs. Against this background, LROE and LACEs seem equally suitable. LACEs require simplification in price calculation. This paper attempts to provide the price time series needed to calculate the LROE using the forecasting model described below.

## 1.3 Existing electricity price forecasting models

Extensive literature is available in the field of modelling and forecasting electricity exchange prices. However, there is no consensus on the approach and methodology of modelling. Most models are designed for a specific market situation or forecasting horizon in which they perform well and deliver robust results. In an extensive review, Weron has identified five different categories of electricity price forecasting models: Multi-agent, Fundamental, Reduced-form, Statistical and Computational intelligence models. Hybrid combinations of model types do also exist (Weron, 2014). This classification will be used for assessing the state-of-the-art forecasting models as well as the newly developed model in this paper. In the following, some practical examples of forecasting models are presented which are of methodological interest for this work.

In 2010 Jonsson et al. investigate the influence of wind energy forecasts and actual wind volume on the Danish electricity exchange price. They use a non-parametric regression model and a statistical distribution of the spot price for different scenarios and conclude that a high forecast feed-in from wind turbines lowers the electricity price. The actual amount of wind energy also influences the price. The authors observe growing price volatility and weather-related price patterns (Jónsson et al., 2010). Jonsson et al. focus on the impact of wind energy onto the electricity price, but do not consider the impact of the electricity price during different wind conditions onto the economic efficiency of WTs. The correlation between actually available wind energy and electricity exchange price as described by Jonsson et al. should be considered.

In 2013 Jonsson et al. then pursue a two-step approach to model the short-term spot prices in Denmark for the years 2010 and 2011. They forecast grid load and electricity generation from WTs. Non-linear and transient influences of these two variables are considered in the first step of the model by a non-parametric regression. Subsequently, time series-based models are used to represent remaining autocorrelations and seasonal effects. The authors conclude that models with variable parameter estimation can yield better results over time than those with static parameters (Jonsson et al., 2013). However, robust parameter estimation has the advantage that models are less vulnerable to abrupt parameter changes e.g. due to excessive price peaks. Hence, robust parameter estimation seems to be preferable for the given case.

Fanone et al. can generate both negative and positive price peaks with a parameter rich fundamental forecasting model of the German intraday market with hourly resolution. The model parameters are calibrated using historical EPEX intraday data. The hourly spot price is divided into two components, namely a time-dependent adjustable component and a deterministic component containing long-term variations and seasonal effects. An annual and a half-yearly periodicity can be observed when investigating daily spot prices (Fanone et al., 2013). A disadvantage of this approach is that fundamental changes in generation capacities such as decommissioning of coal powered plants are neglected. This could lead to long-term forecasting errors that could be avoided by introducing these changes in generation capacities.

Šumbera and Dlouhý model the German spot market based on the fundamental assumption that the demand for electricity and the system load always equal the generation capacity provided by all power plants. A merit-order approach is used for pricing. The power plants are divided into dispatchable and non-dispatchable power plants. The dispatchable power plant schedules are presented in high detail. Non-dispatchable power plants are grouped according to their energy source and defined as "must-

runs". Power plants whose generation depends only on their availability are modelled with variable costs of zero (Šumbera and Dlouhý, 2015). A disadvantage of this methodology is that a set of all generation units or at least a representative data set must be available. Apart from that, the merit-order approach as chosen by the authors seems to be promising for this research.

Next to the discussed models that specifically address forecasting electricity prices there is also a broad variety of equilibrium
models that analyze energy systems and consequently may be used for electricity price estimation as well. The Balmorel model will be discussed as representative for this group. Balmorel is a partial equilibrium model for analyzing the electricity and combined heat and power sectors in a large geographical and international perspective. It has been directed towards the solution of an optimization problem in GAMS to determine entities like generation, consumption, transmission and prices of electricity and heat as well as emission. The source code of Balmorel is openly available since 2001 (Wiese et al., 2018). Due to the wide
range of applications and the necessity to solve an optimization problem, the model poses comparatively high requirements in terms of of technological detail and data, even if these can be reduced by later model adjustments.

Another widely used, commercial tool is the market simulation software PLEXOS (Papadopoulos et al.). Like Balmorel, PLEXOS is able to solve complex optimization problems with an object-oriented approach and delivers market results for the gas, electricity and water system. Thanks to parallel computing, PLEXOS can boast comparatively short calculation times in
the range of a few hours up to minutes (Energy Exemplar, 2019). Yet, it is questionable whether a leaner model might not be sufficient to achieve comparative results at lower computational effort.

In the following, a new price forecasting model is derived that combines the advantages of the above approaches to predict time series data of the electricity exchange price in a lean way.

**2 Methodology and forecasting model**

In order to draw conclusions on the economic perspective of wind turbines in Germany, model requirements are derived from the previous chapters. Based on Section 1.1, the forecasting objective of this study should be the day-ahead spot market as it is one of the most relevant markets for trading electricity from wind turbines and furthermore functions as a reference for drafting PPAs. On the day-ahead market electricity is traded for each hour of the following day.

To assess the potential revenue of WTs, the entire life cycle has to be considered. Therefore, the long-term development of the
electricity exchange price should be estimated. At the same time, increased price volatility due to higher generation from renewable sources should be considered by the chosen temporal resolution. In conclusion, a long-term forecast period of 20 years at a high temporal resolution will be modelled to fulfill both requirements. Due to the hourly trading steps of the day-ahead market, the model is designed to calculate prices in hourly resolution to ensure comparability.

Calculating hourly prices requires that short-term demand characteristics like daily and weekly patterns are modeled. Seasonal
demand variations as well as long term developments of the average demand should be possible to include. For this study it is assumed that the hourly demand for the entire forecast period is known in foresight and that the hourly demand equals the grid load at all times.

The available generation capacity from renewable and conventional sources needs to be considered by generation technology. It is also required, that changes in national electricity generation landscapes can be parametrized in the model to account for developments like the decommissioning of coal and nuclear energy in Germany. Electricity from weather dependent renewable sources must be integrated. Finally, neighboring electricity markets should be considered in terms of electricity import and export.

A general criticism of many existing models is the lack of transparency and accessibility of the calculation methods and databases as they are often not openly accessible. Therefore, only openly accessible data shall be used for the forecasting model.

## 2.1 Forecasting objective and model classification

The subject of this study is the hourly German day-ahead spot market price over the next twenty years. The developed forecast model uses a merit-order approach to calculate hourly prices and can therefore basically be classified as a fundamental model according to Weron (Weron, 2014). An object-oriented approach has been added for the implementation of power plants. Each conventional electricity generation unit functions as a single agent and can be parametrized individually. Due to this design decision, the model can moreover be assigned as an agent-based model, yielding additional benefits over a solely fundamental procedure. All in all, according to Weron, the model presented is being classified as a hybrid model, combining fundamental and agent-based aspects. Figure 1 shows which input variables and calculations are necessary to describe this target and how they have been linked together within the presented model. First, the electricity demand is calculated for every hour of the forecasting period based on a mean annual value and an hourly fluctuation factor. Afterwards, marginal generation cost and generation capacities of conventional power plants are derived from different operational parameters and commodity prices. The hourly renewable generation capacity from solar and wind energy is calculated based on installed capacity and an hourly generation potential. The hourly electricity price is derived as the marginal generation cost of the most expensive power plant that is still needed to meet the current electricity demand. The obtained time series data are used to calculate LROE for economic evaluation of wind turbines.

In the following Sections the input data and intermediate steps of the calculation model are explained in more detail.

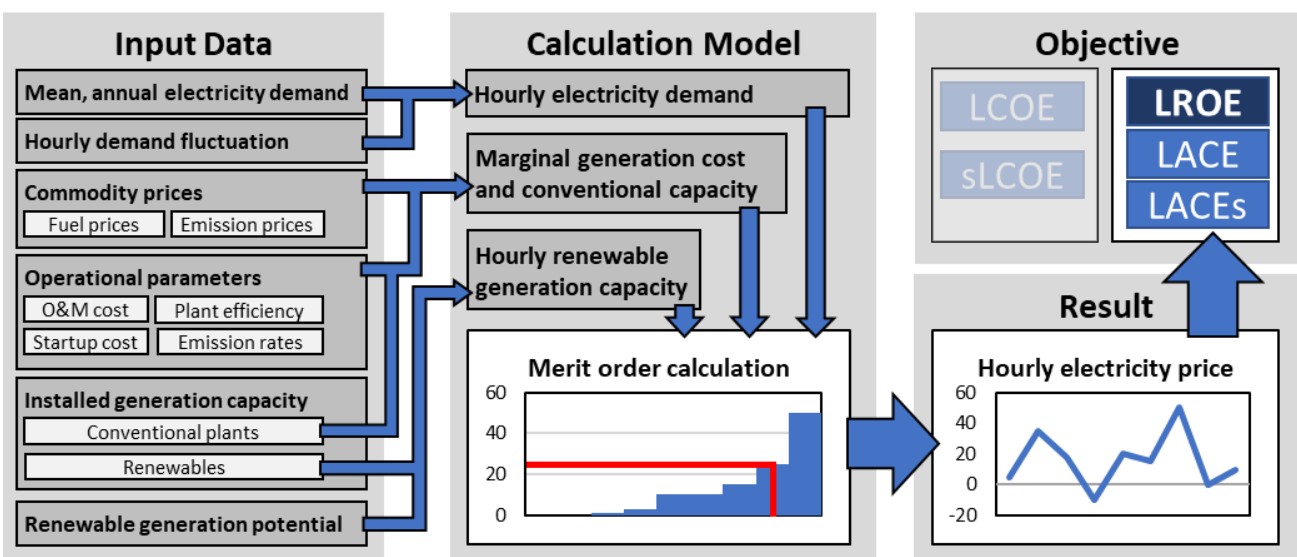

**Figure 1: Schematic model structure and input data requirements**

### 2.2 Hourly electricity demand

The hourly electricity demand of a country for each year is composed of an annual mean value $D_{mean}$ and an hourly fluctuation factor $f_{var}$ according to Eq. (1). $D_{mean}$ is derived from the total annual demand required as model input while $f_{var}$ remains the same for any scenario. The time series for $f_{var}$ is derived from data of the Ten Year Network Development Plan (TYNDP18) of the European network of transmission system operators (entso-e, 2018). Figure 2 shows annual, weekly and daily sections of $f_{var}$. It can be observed how $f_{var}$ covers different cyclic characteristics of the actual electricity demand like higher demands

during winter as well as peak and off-peak hours. All long-term demand trends must be considered within $D_{mean}$. The hourly electricity demand is assumed to be equivalent to the load profile used for the merit-order approach.

$$D(t) = D_{mean} \cdot (1 + f_{var}(t)) \tag{1}$$

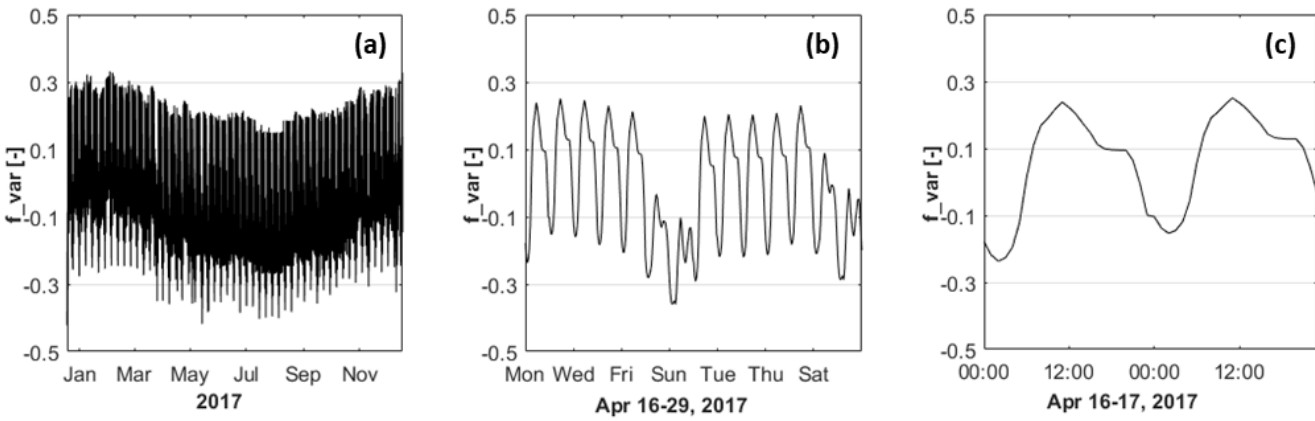

Figure 2: Demand variation factor $f_{var}$ for Germany in 2017 with annual (a), weekly (b) and daily (c) characteristics

## 2.3 Marginal generation cost and conventional capacity

The national installed generation capacities are divided into conventional and renewable plants, where gas, hard coal, lignite, oil and nuclear fueled plants are considered as conventional. On the other hand, hydro, solar, wind (onshore and offshore) and others (mainly biomass) are considered as renewable. Based on these categories, a total installed capacity per year is required as user-input. In a next step the model derives individual objects to generate an object-oriented plant fleet based on reference data. The procedure can be classified as a simple agent-based approach. The number of power plants constitutes the number of agents. Because of the underlying merit-order approach, the decision-making heuristic for every agent and hour is to offer electricity at their own marginal generation cost and to generate and sell electricity if they are below the uniform market clearing price. An interaction topology is given by the competition to the other power plants or agents. The exogenously given electricity demand represents the environmental influence that drives the decision of every agent.

Marginal generation cost $c_{var}$ is calculated for every conventional plant following Eq. (2)

$$c_{var} = c_{fuel} + c_{o\&m} + c_{CO_2} \qquad \text{with} \qquad c_{fuel} = \frac{p_{fuel}}{\eta} \qquad \text{and} \qquad c_{co_2} = \frac{p_{CO_2}}{\eta} \cdot f_{co_2} \qquad (2)$$

where $c_{var}$ is the marginal generation cost of a specific power plant used for the merit-order approach. Commodity prices are split into fuel prices $p_{fuel}$ and emission ($CO_2$) prices $p_{CO_2}$ which are both assumed to be constant over time during a simulation run. The resulting cost are then calculated regarding both, efficiency $\eta$ of plants as well as emission rate $f_{CO_2}$.

For this research, values given in Table 2 have been assumed as reference for commodity prices and efficiencies. The values have been derived from TYNDP18 data and are assumed to be constant over time (entso-e, 2018). The specific cost terms can be varied for each individual power plant. When adding additional plants, cost values can be set individually. For the plant efficiency it was assumed that over all power plants the efficiency follows a beta-distribution defined by $\eta_{min}$, $\eta_{mean}$ and $\eta_{max}$ where the oldest plants operate at the lowest efficiency and vice versa. Every plant is also given a date of commission and

shutdown date. Outside the resulting time span, the respective power plant is not considered for the price calculation. The given emission factors refer exclusively to emissions occurring during operation.

Table 2: Reference values for commodity prices and plant efficiencies (entso-e, 2018)

| Property | Sign | Unit | Gas | Hard Coal | Lignite | Oil | Nuclear |
|---|---|---|---|---|---|---|---|
| O&M cost | $c_{o\&m}$ | €/MWh | 1.46 | 3.3 | 3.3 | 2.57 | 9 |
| min efficiency | $\eta_{min}$ | % | 25 | 30 | 30 | 25 | 30 |
| mean efficieny | $\eta_{mean}$ | % | 44 | 40 | 40 | 36 | 33 |
| max efficiency | $\eta_{max}$ | % | 60 | 46 | 46 | 43 | 35 |
| fuel price | $p_{fuel}$ | €/MWh | 21.96 | 8.28 | 3.96 | 50.76 | 1.69 |
| emission rate | $f_{CO_2}$ | kg/MWh | 205.2 | 338.4 | 363.6 | 280.8 | 0 |

## 2.4 Implementation of cross-border transactions

When considering the German market, the import and export of electricity is very relevant due to its many neighboring
countries and its central location in the increasingly interconnected European power grid. The approach for this paper on integrating cross-border transactions is to model neighboring countries as single power plants (agents). These agents are assigned individual capacity and dynamic marginal cost so that they can be included into the merit-order plot. The net transfer capacity (NTC) provided by the European network of transmission system operators is assumed to be the technical upper bound for cross-border electricity transfer. Regarding the merit-order plot, this corresponds to the capacity (bar width) of the
agent. NTC values for Germany are implemented as given in Table 3. At this state, NTC is assumed to be constant over time for all countries.

Table 3: Cross-border NTC capacities for Germany (entso-e, 2018)

| Country | AT | BE | CH | CZ | DK | FR | LU | NL | NO | PL | SE |
|---|---|---|---|---|---|---|---|---|---|---|---|
| NTC [MW] | 5000 | 1000 | 4600 | 2100 | 2765 | 1800 | 2300 | 4250 | 1400 | 2500 | 615 |

In case of the neighboring countries the NTC can function as both, demand capacity and supply capacity, depending on the current electricity spot price of the country. The spot price is set as the marginal cost (bar height) for neighboring countries.
The bar height determines whether there is electricity import or export at a certain hour. If the local price of a neighboring country is lower than the German price, it is assumed that Germany will import electricity from this country to the extent of available NTC. The neighboring country thereby acts as a supplying power plant. On the other hand, if the local price in a neighboring country is higher than the German price, it is assumed that Germany will export electricity to the extent of available NTC. In this case the available NTC enlarges the current electricity demand. To estimate hourly prices for the neighboring
markets a simplified pre-simulation for each country is executed where import and export are neglected. This pre-simulation

is based on the current plant portfolio and annual demand of each neighboring country and thereby also respects its generation mix.

Figure 3(a) illustrates this approach for one hour within the model. The electricity demand on the German market is depicted by the dotted line at 80 GW. Whenever the market price on a neighboring market is lower than the local price, the neighboring market is handled as a power plant that supplies electricity for the German market. This applies to the two countries at the left side of the dotted line. When a country exhibits a higher market price than the local price it is assumed to be a potential market for export of electricity. This means that the actual demand can be extended by the NTC of the according countries. The continuous line at 105 GW shows the resulting total demand that finally defines the price. In this case the German spot price increases due to cross-border transactions, because there are several markets available for export. Figure 3(b) shows an exemplary weekly course of the modelled cross-border transfer. It can be seen that at all times there is import and export at the same time, while in the overall balance there is more export for this particular week.

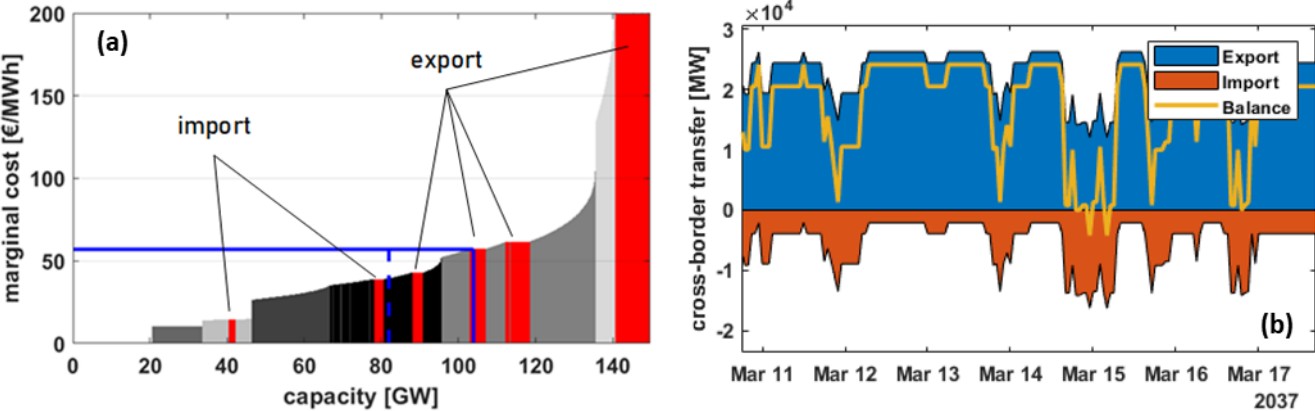

**Figure 3: Exemplary merit-order plot from forecasting model with cross-border considerations (a) and exemplary weekly course of modelled cross-border transfer (b)**

## 2.5 Hourly renewable generation capacity and weather time series data

For the implementation of weather-dependent electricity generation technologies such as photovoltaic and WTs, the underlying weather time series data are of fundamental importance. In this paper, the influences of wind speed and solar radiation are assumed according to previous studies of Staffell and Pfenninger (Staffell and Pfenninger, 2016; Pfenninger and Staffell, 2016). The authors use weather data from global reanalysis models and satellite observations to generate synchronized national time series data for solar and wind generation capacity factors for the years 1985 to 2016 at an hourly resolution. This data is also used for commercial application in the dataset PLEXOS World 2015 (Brinkerink and Deane, 2020).

It is assumed that there is no long-term weather trend in terms of wind and solar radiation. To calculate the hourly available electricity generation, the technology specific capacity factors are multiplied with the overall installed generation capacity of solar panels and WTs. In addition, the assumption is made that capacity factors will not change in the future. This poses an

underestimation of the available electricity because an increase in capacity factors may be expected due to technological progress.

The influence of the ambient temperature on the electricity demand and spot price was assessed by investigating historic data of the German day-ahead spot price, trading volume and temperature in Germany. Hourly data from 2012 to 2017 has been used and analyzed in terms of correlation (Deutscher Wetterdienst, 2020). Figure 4 shows the results of this analysis in scatter

plots with a least square regression line and correlation coefficient $r$. It can be seen that there is a weak positive correlation ($r = 0,1652$) between temperature and trading volume and a weak negative correlation between temperature and spot price ($r = -0,1270$) as well as trading volume and spot price ($r = -0,1454$). For the given model it is assumed that the electricity demand always corresponds exactly to the trading volume. Under this condition and given the very low correlation coefficients, the assumption is made that weather data and load profile do not need to be synchronized further for the given case. All other

weather and climate influences are also neglected within this study.

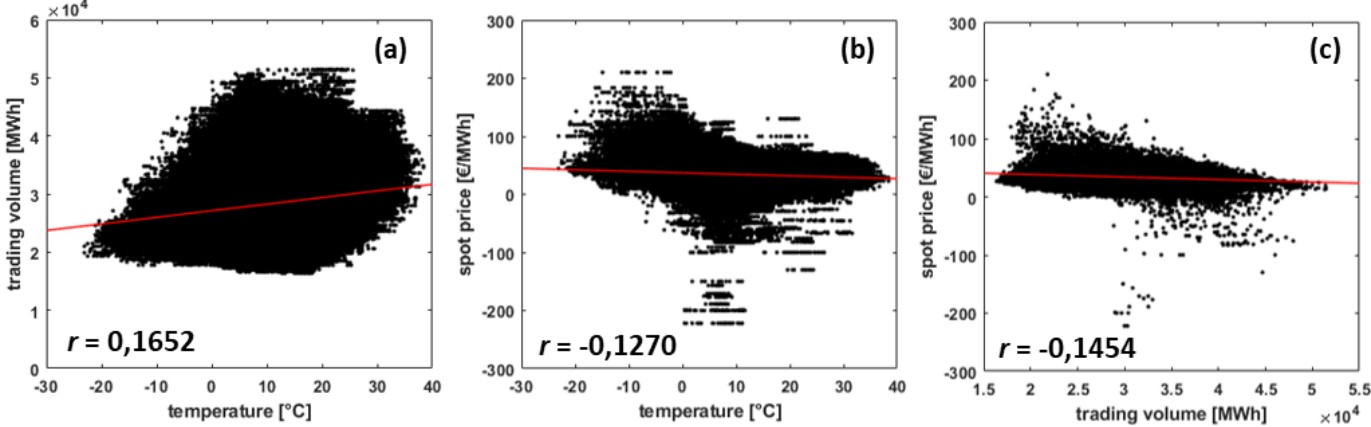

**Figure 4: Scatter plots, least square regression line and correlation coefficient $r$ for temperature and trading volume (a) temperature and spot price (b) and trading volume and spot price(c) for the German day-ahead market from 2012 to 2017 at hourly resolution**

**2.6 Model validation**

To validate the model, calculated prices have been compared to actual prices on the EPEX Spot day-ahead market for the year 2017. As Germany and Austria have been a coupled market until 2018, the test has been executed for both countries together. Figure 5(a) shows the ordered annual price duration curve for Germany and Austria as well as the prices calculated by the presented forecasting model. It can be stated that the model provides satisfactory results in reconstructing the average price

level and price distribution at a mean absolute error of 2.38 €/MWh and root mean square error of 5,8 €/MWh over the course of a year. The mean spot price varies by 1,87% between historic data (34,2 €/MWh) and model result (34,84 €/MWh). Anyhow, it can also be seen in Figure 5(b) that the model is not able to fit the actual temporal order of spot prices. Especially the number of negative price peaks during summer is overestimated, leading to an overall RMSE of 20,26 €/MWh between historic and calculated values. Figure 5(c) underlines, that this is the case all over the year. The daily RMSE is in the range of

10 – 20 €/MWh most of the time. In conclusion, the given model results are sufficient when regarding the overall price structure over a year but must not be used for the forecast of exact hourly values.

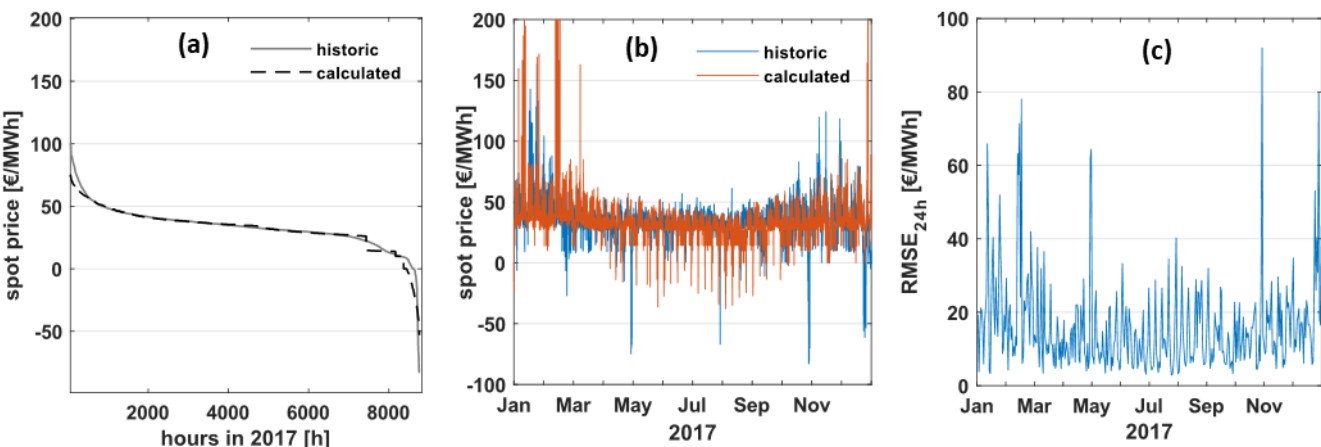

**Figure 5: Validation results, comparison of model results against historic values from EPEX Spot for Germany in 2017**

The average calculation time for a forecast period of 22 years at an hourly resolution lies at 00:05:11 on a regular home
computer without parallelization. In comparison, PLEXOS calculates optimization results for comparable time periods and resolution in several hours, even with parallel computing (Energy Exemplar, 2019).

## 3 Model application, results and case study

An important question which this research tries to address is, how do different future expansion scenarios influence revenues of WTs. Therefore, the forecasting model described in Section 2 is now used to calculate hourly electricity prices for the
German market over the course of the next twenty years. Calculation results will be analyzed in Section 3.1 and finally put into perspective in a brief case study in Section 3.2. For this study, two different renewable expansion scenarios based on German legislation and policy goals are being evaluated for the years 2019 to 2040. Both scenarios are based on the requirements of the Renewable Energy Act and assume that the medium and long-term energy policy objectives of the German government will largely be met. Scenario A represents the increase in the share of electricity generated from renewable
energies in gross electricity consumption to 65 % by 2030 as stipulated in the coalition agreement of March 12, 2018 (CDU et al., 2018). Conventional generation capacities are assumed to remain constant in this scenario. Scenario B is derived from the scenario framework approved by the German Federal Network Agency in June 2018 (Bundesnetzagentur, 2018). In addition to scenario A, it includes in particular the phasing out of nuclear power by 2022 and of coal powered plants until 2038 decided by the German Coal Commission in 2019. The forecast lignite and hard coal-fired power plant capacities are based on
standardized assumptions on the technical and economic lifetime of power plants. The chosen scenario B follows the basic idea of a moderate sector coupling and a mix of centralized and decentralized structures. It forecasts the development up to

2035. Thereafter, the forecast for gas powered plants and renewable sources is linearly extrapolated for the following five years.

In Figure 6 the overall installed capacity for Germany is shown for Scenario A (renewable energy expansion pursuant to the statutory expansion path of EEG 2017) and Scenario B (additional dismantling of coal and nuclear plants). Mean annual demand is assumed to be constant for both scenarios.

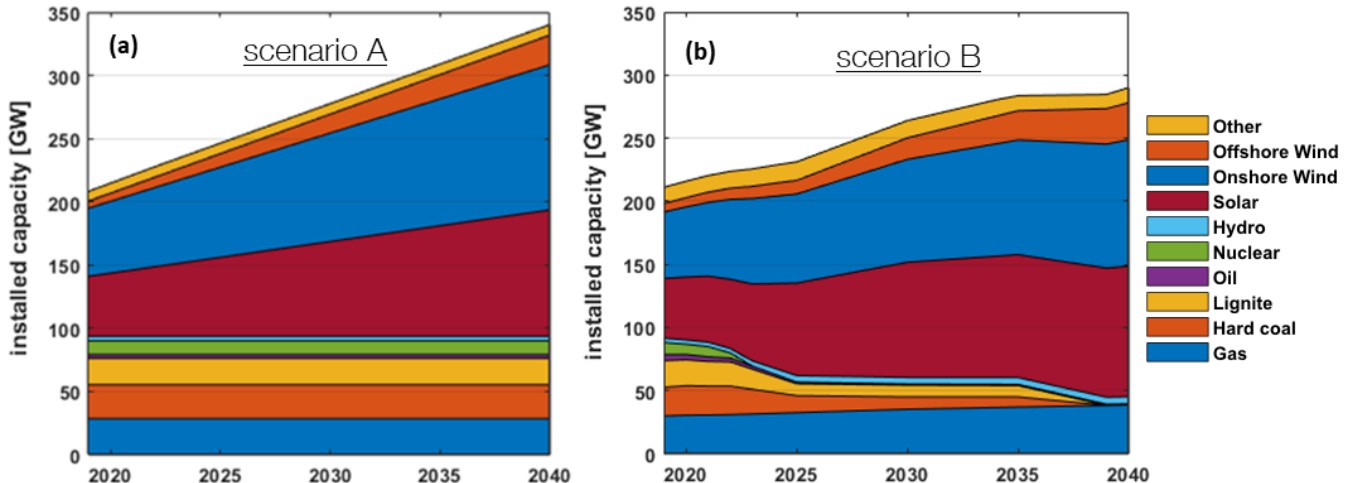

**Figure 6: Development of installed capacity in Germany pursuant to statutory expansion (a) and additional dismantling of conventional plants (b)**

It should be emphasized that Scenario A with the pure addition of renewables is a highly improbable scenario. For this study it is used as a basis for scenario B and to indicate the isolated influence of the renewable energy expansion on the German spot market price.

## 3.1 Scenario results and analysis

Figures 7(a)-(c) show the impact of increasing electricity generation from renewables on the average spot price and price volatility for the given scenarios A and B. Price volatility is represented in Figure 7(c) by the floating standard deviation of spot prices at a window size of 365 days.

Both scenarios use the same time series data for annual generation from renewables as shown in Figure 7(a). Comparing Figure 7(a) and (b), it is visible for scenario A that the increasing generation from renewables causes a constant decrease of the spot market price. This is referred to as the merit-order effect (Sensfuß et al., 2008). At the same time an increasing price volatility can be observed in Figure 7(c). For scenario B it can be seen in Figure 7(b) how the decommissioning of conventional power plants counteracts the merit-order effect seen for scenario A. Rising prices can be observed for the next five years due to the phase-out of nuclear energy. Prices then fall until 2035 along the renewable energy expansion and finally rise again with the complete decommissioning of coal energy. In this case the average price level remains more or less at a constant level. On the other hand, a more strongly increasing price volatility for scenario B can be observed in Figure 7(c). Taking the year 2035

as an example, it is clear to see how reduced generation from renewable sources leads to an increase in price and decreasing price volatility. In reality, this could be the case e.g. in a weak wind year.

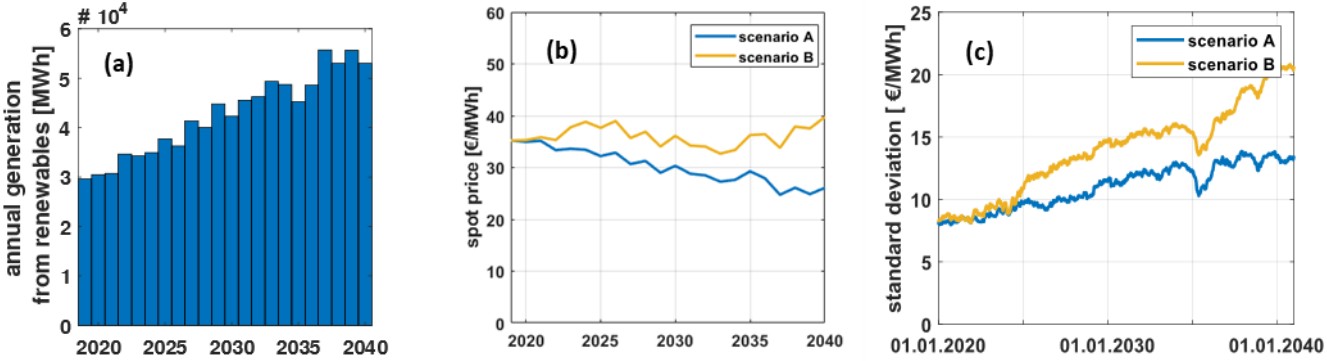

**Figure 7: Influence of rising feed from renewables (a) on annual mean spot price (b) and price standard deviation (c) at otherwise constant conditions**

A sensitivity analysis for the $CO_2$ price is investigated on top of both scenarios A and B. Four different specific prices are analyzed, namely 10, 18, 30 and 60 €/t. Figure 8(a) and (b) show the corresponding results and sensitivity of spot prices against the $CO_2$ price. It can be observed how an increased emission price leads to higher mean spot prices. This influence becomes stronger the more conventional plants are active on the market. The converging lines in Figure 8(b) along the expansion shown in Figure 6(b) emphasize this relation. A minimal taper of the curves is also shown in Figure 8(a). This is due to the fact that

even without the decommissioning scheme, renewable energies are pushing conventional power plants out of the market.

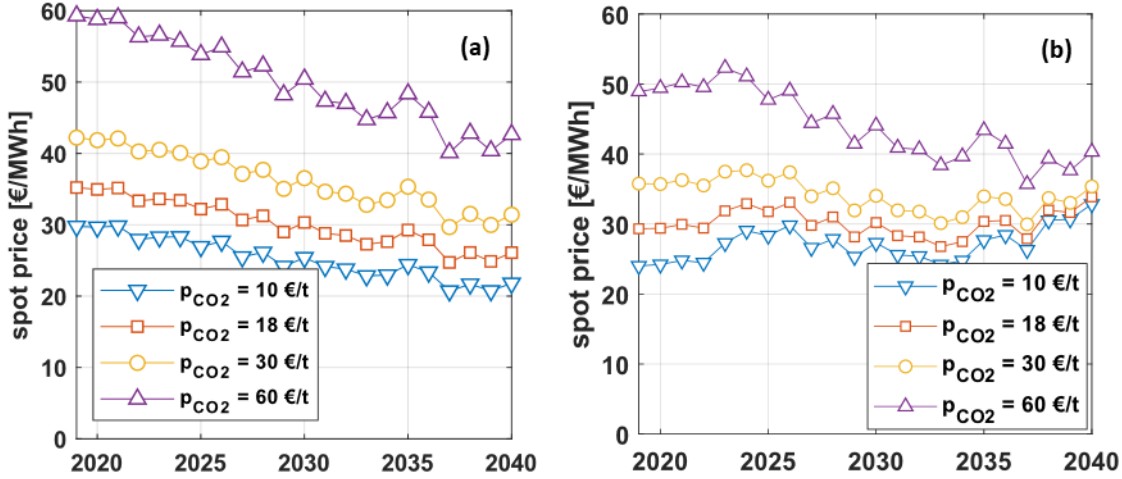

**Figure 8: Mean annual spot price in Germany for scenario A and B at different $CO_2$ prices**

### 3.2 Case study observing a small onshore wind park in Germany

The model results are used to evaluate the potential revenue of WTs in a brief case study. Therefore, hourly SCADA data of a
German wind park with 5 turbines of the 3 MW class with average full load hours of 1920 h/a was used. Following Section 1.2, the commonly used LCOE is no holistic measure to assess revenue potential and overall profitability of WTs. Instead the LROE as introduced by Baker in 2011 will be used. The modelled hourly prices from the merit order approach are used to determine the time course of the revenues needed to calculate the LROE.

Revenues from selling electricity are calculated based on the extrapolated SCADA data and the modelled spot price over the
course of the next twenty years. Figure 9 shows the LROE of the investigated wind farm at two different emission prices (18 and 60 €/MWh) for both scenarios. These results are compared to potential revenues based on hypothetic PPAs with different base prices. Mendicino et al. propose that feed-in tariffs of corporate PPAs over 7 to 10 years should be in the range between 75 €/MWh and 100 €/MWh (Mendicino et al., 2019). For this study a lower range of base feed-in tariffs is assumed because of the longer time span (40 €/MWh – 50 €/MWh). In the constructed case, electricity sold during times of spot prices below
the base price will be remunerated with the respective base PPA price. In order to assess the profitability, the results are compared to current estimates of the LCOE of onshore wind turbines in Germany (Kost et al., 2018; IWR Online, 2019; Wallasch et al., 2019). These are shown in Figure 9 as green box plots.

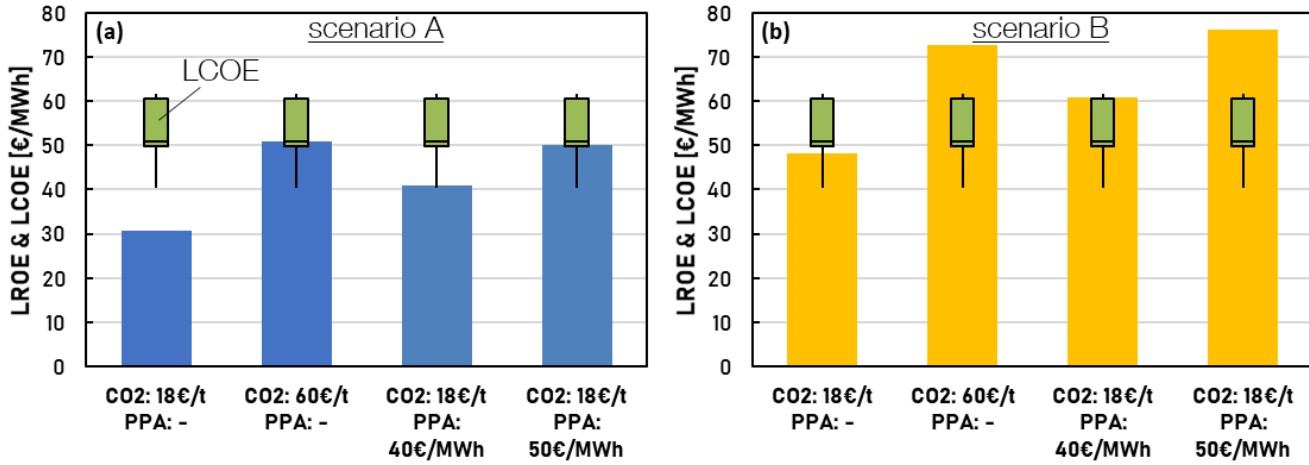

**Figure 9: LROE of the observed wind park over 22 years at different emission prices and different PPAs compared to currently**
**estimated LCOE of onshore WT in Germany (green box plots) for expansion scenario A (a) scenario B (b)**

On the one hand, it can be seen that in the event of higher electricity exchange prices, higher revenues can be expected for all cases. Furthermore, it can be observed that for this particular case, the chosen PPAs do not necessarily guarantee profitability. However, the lower revenues from PPAs should always be evaluated against the background of the equally lower risk.

For the results as shown in Figure 9, many turbines will not be profitable without subsidies for all cases in scenario A and for
scenario B at emission prices of 18 €/t without PPA. For these cases the range of LCOE lies mostly above the LROE. For the other three situations most or all turbines would be economically profitable.

Based on the case study, it can be seen that profitable operation of wind turbines in Germany without subsidies is uncertain in the future. Depending on which scenario is assumed and which way of selling is chosen, future plants could be uneconomical. On the one hand, this emphasizes the need to further reduce the LCOE of wind energy through technical progress in design, production and operation. However, this finding may also lead to various developments in order to ensure the economic viability of wind turbines in Germany after all. For instance, it could lead to rethinking existing remuneration and auction systems, also to prevent cannibalistic merit-order effects. Another possibility would be to consider further governmental support in legislation or adjustment of emission prices. Finally, an increased market share of PPAs could be the consequence in the future. However, these would have to be set at a sufficiently high price.

## 3.3 Conclusion

The model application and corresponding results show that it is possible to forecast electricity exchange prices with the presented model at comparatively low data requirements and computational costs. The model results can be used for the derivation of development goals in terms of LCOE and deliver the necessary values for break-even consideration in terms of cost reduction or annual energy production.

Influences of renewable energy expansion and the decommissioning of conventional power plants on the mean spot price can be shown in two calculation scenarios in Section 3.1. The model results in Figure 7(a)-(c) show that for scenario A, a pure expansion of renewables could lower the electricity price in Germany by 10,98 €/MWh (-31,2 % compared to 2019) and increase its standard deviation hence volatility by 5,36 €/MWh (+67 % compared to 2019). In addition, results for scenario B show that the forecasted expansion of renewables in Germany, in conjunction with the coal and nuclear power phase-out, can lead to roughly constant exchange market prices and increasing volatility by up to 12,37 €/MWh (+151 % compared to 2019). Figure 8(a) and (b) show that the pricing of emissions in the coming years will have a strong influence on the exchange price. This effect will decrease as more and more conventional plants are being decommissioned. Overall, the level of emission prices in the next 20 years has a very strong influence on both the exchange price and the profitability of non-subsidized WTs. Figure 9 shows that at a $CO_2$ price of 18€/t most of the onshore wind turbines in Germany could not be operated without additional funding. Regarding the evaluation of revenue potential, LROE, as presented in Section 3, has shown to be an appropriate benchmark for evaluating market developments. By using LROE instead of LCOE model results can be evaluated independently from plant specific cost and thereby have a more general character and applicability. Just like LCOE, LROE can be used to define and evaluate technical and financial development goals for engineering. Moreover, it allows a consideration detached from plant costs and can be used both in the negotiation of PPAs or as a benchmark for policymaking, for example in determining a suitable $CO_2$ price. Today, subsidies in the form of the tendering procedure generally follow the LCOE. A funding which considers the LROE for different technologies could be a more holistic approach and a more indirect technology support.

## 4 Discussion and outlook

In this study a new forecasting model has been presented that estimates future electricity exchange prices for Germany in order to conclude on potential revenues of non-subsidized WTs. Prices are calculated at an hourly resolution over 22 years. Historically, this used to be a rather unusual combination. However, the necessity to consider the dynamic electricity generation characteristics of wind and solar energy has become more common in recent years and state-of-the-art models as described in Section 1.3. The given model is using a merit-order approach in combination with a simple multi-agent approach for conventional power plants. The latter allows to integrate neighboring countries and cross-border electricity trading. The developed model is constructed comparatively simple with many assumptions being made. This leads to low data requirements based on open source data and allows easy adaptation on the one hand. Compared to modern complex optimization models this may be advantageous. On the other hand, the model assumptions also cause less accurate results and narrow the possible field of application.

To validate the model results, historic prices of the German day-ahead market of 2017 have been compared to model results for the same year. The annual ordered price duration curve was reconstructed at a mean absolute error of 2.38 €/MWh. Anyhow, because of the simplifying assumptions regarding electricity demand and weather data synchronization it is not possible to forecast the exact temporal course of prices by the hour. For future work it is planned to benchmark the given model results in terms of accuracy against state-of-the-art models like PLEXOS or Balmorel. Even if the model results lag behind proprietary solvers in terms of result quality, the results can at least serve as a first estimation and comparison value that can be generated within a few minutes.

The two scenarios discussed in this study are solely developed from current German policy goals. Especially scenario A is very unlikely to actually happen. Further, more sophisticated expansion scenarios for Germany and other European countries that also consider long-term electricity demand trends should be simulated. The scenarios from the IEA World Energy Outlook and the Ten Year Network Development Plan 2020 by ENTSO-E are currently being considered for this purpose.

A major limitation of the model lies in the neglect of national grid capacities. Grid bottlenecks are already posing major challenges for the expansion of renewable energies today, for example when considering the integration of offshore wind energy. This strong simplification should be improved in subsequent model extensions. The given model also needs further improvement to overcome current limitations. During future studies, the model shall be extended by implementing dynamic time series for the currently constant parameters like emission and fuel prices as well as cross-border capacity and average electricity demand. Also finding rules for synchronizing the weather data used and the electricity demand time series might yield further improvement of the model results. Finally, the agent-based approach could be further developed by introducing randomness as well as learning rules for agent decision making. Further model application is also planned in combination with planning and optimization tools such as the wind farm optimizer WIFO to generate a more reliable economic yield prognosis in addition to the energy yield prognosis. WIFO is an optimization tool that calculates wind farm layouts based on LCOE minimization and maximization of annual energy production (Roscher et al., 2018; Roscher, 2020).

## Acknowledgements

We thank the German Federal Ministry for Economic Affairs and Energy for funding the research project ArkESE in which we address this topic. In the three-year project (start December 2018), rural municipalities are investigated to design individual electricity supply systems on the basis of renewables. Municipalities shall be enabled to prepare, make and implement energy system related decisions more self-sufficiently.

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
