# Peer review of "Future Economic Perspective and Potential Revenue of Non-Subsidized Wind Turbines in Germany"

_Wind Energy Science, 2020_

## Referee Comment (RC1) · Anonymous Referee #1 · 2 Apr 2020

Thank you for this nice publication. The paper presents a possibility to forecast the development of energy spot prices with respect to expansion of renewable energies. The paper clearly states the benefit of renewables by using an understandable method.

- Nonetheless the paper is missing a clear conclusion. At the moment, the conclusion is mixed with the discussion and outlook.

- The scenario selection should be more elaborated and discussed. What would be if the expansion of renewables is decreased overtime. As Space and possibilities are limited. What is the likelihood of both scenarios?

- Section 2.5 should be started with the idea that cross-border interaction are modelled

by as power plants. However, what happens if the neighboring countries are demanding instead of delivering?

- In Section 2.6, why did the model not test on a single country? You mentioned the markets have been separated in 2018.

- I personally have my doubts that you are the first one to think of LROE. I assume the citation is missing here.

- Rephrasing of line 220, the content does not come across.

- Figure 7a: what is the Co2 price for these graphs?

Overall, the paper is missing some comma and might be rephrased at certain points. Examples: - have to = must - Finally, ... - By forecasting future prices, it is ... - table design is not consistent

---

## Referee Comment (RC2) · Matti Koivisto (Referee) · 8 Apr 2020

In the paper "Future Economic Perspective and Potential Revenue of Non-Subsidized Wind Turbines in Germany", the authors present a model for estimating future spot market prices in Germany. Two different scenarios towards 2040 are then presented, considering also different CO2 prices. Revenue from a case study wind power plant is compared for different CO2 price scenarios, with a different PPA purchase prices also considered. The paper presents a very important analysis, considering the possible challenges of non-subsidized wind power plants in the future being able to generate sufficient revenue. However, I ask the authors to go through the comments below. The

authors should clarify especially the LROE concept, and provide clearer conclusions of the presented analyses (these are the two final comments).

1) The authors provide a comparison of the presented electricity price model to literature, but focus mainly on regression/time series type modelling. However, there are methods similar to the presented model implemented in energy system modelling tools, such as PLEXOS (https://energyexemplar.com/solutions/plexos/) or Balmorel (http://www.balmorel.com/; code freely available). It would be beneficial if the authors could expand the literature review to one/some of the energy system modelling tools (some open tools: https://en.wikipedia.org/wiki/Open_energy_system_models).

2) The authors should clarify how the hourly profiles of load and wind and solar generation are modelled towards 2040: 2.a) Wind and solar: In section 3.1., it is said "The annual fluctuations are merely due to different weather conditions in the individual years". But it is not clear to me what weather conditions are assumed, e.g., for 2035 or 2040 (this is not clear from section 2.4). Please clarify (and consider expanding section 2.4). 2.b) Are any changes in the capacity factors of wind and solar generations considered towards 2040, as additional installations (presumably with different technologies, e.g., hub height and turbine type) appear to the system? 2.c) How is load profile modelled for the different years towards 2040? Is the same profile (i.e., time series shape over the year) assumed all the way to 2040, with only the annual energy level changing? Please clarify. 2.d) Are wind, solar and load profiles synchronized? I.e., is for example 2035 defined so that the wind, solar and load data are based on the same weather year (as they all might be correlated due to weather dependencies)?

3) How is annual energy consumption assumed to change towards 2040? As this is a quite fundamental variable for the system, its progression towards 2040 would be nice to see as a figure or table.

4) Considering consumption changes, are, e.g., power-to-gas, electrification of heating and/or electric vehicles considering going towards 2040?

[Figure]

5) Are cross-border capacities (Table 3) assumed to remain on the 2018 level all the way to 2040? Is this justifiable?

6) In section 2.5, it reads "To estimate power import and export every neighboring country is modelled as a hypothetic power plant with individual capacity and marginal cost". Please elaborate a little more on this. E.g., how are the very different generation mixtures in the different countries taken into account (hydro in NO, nuclear in FR, and so on)?

7) About Figure 4: It seems that the model cannot capture the likelihoods of the highest spot prices (i.e., the lines diverge going to the left). Can any discussion be given on to why this happens?

8) Word "marketing revenue" is used in many places. I find it a little bit confusing, as "marketing" usually refers to advertisement. Perhaps "market revenue" could be used? (please disregard this comment if "marketing revenue" is an often used term in this context)

9) Understanding Figure 8: The different CO2 prices for the orange bars provide a clear comparison. However, it is not clear to me how the different PPA prices should be considered. How are the 40 €MWh and 50 €MWh linked to the studied expansion scenario B and the different CO2 prices? In the current form, it seems that the 40 €MWh and 50 €MWh are arbitrary prices, and therefore comparisons between direct market revenues and PPAs is difficult.

10) Are transmission bottlenecks inside Germany and resulting possible curtailment of wind generation considered/modelled?

11) Please elaborate more on the LROE concept. Is it based on existing literature? For LCOE, the top part (dividend) are costs, whereas in LROE they are revenue (the divisor seems to be the same in both, namely energy produced). This seems to be a very significant difference (cost = expense vs. revenue = income). Please discuss a

little more on LCOE versus LROE, and why LROE is considered a good measure in the paper. For LCOE, the resulting €MWh value can be understood as the minimum (constant) electricity price over the lifetime to make the project profitable. How resulting LROE values should be understood? (please link the discussion also to the LROE values reported in Figure 8).

12) The Introduction says "A scenario analysis highlights that most of today's wind turbines are not able to yield financial profit over their lifetime without guaranteed subsidies in Germany". However, I don't see this result clearly presented later in the paper. Please provide a clear conclusion section, where each result is presented and explained based on the presented analyses.

---

## Author Comment (AC1) · 20 May 2020

Thank you for your very helpful review.

As for the conclusion, it is agreed that this part is not given enough attention in the current version of the publication. The revised version will contain a more extensive conclusion (Section 3.3) and also discussion regarding the LROE concept.

During this paper it was intended to present and discuss the given model with a first sensitivity study towards emission prices and a brief case study. For this purpose, two simple expansion scenarios have been used where Scenario A only depicts expansion

in accordance with the targets of the renewable energy act and therefore is solely theoretical. Scenario B was given by entso-e results and might be possible to happen in reality. Anyhow, a broadened scenario analysis with further expansion possibilities would not fit the scope of this paper, but rather have the potential for another interesting investigation and downstream application of the given model.

By handling neighboring countries as calculatory power plants within the merit order approach it is indeed possible to display the case that a neighboring country is demanding. This is described in Section 2.5 alongside Figure 3. The revised version of this paper will include some more expansions at this section to enhance comprehensibility.

During the time of this papers emergence there was simply no data for the sole German market available for an entire year (2019). Application on the separated market is intended for future works.

Hardly any existing literature on the concept of LROE was found, except for one forum article which will be cited in the revised version of the paper. Please let us know if and where there is additional scientific literature about this topic that the authors are not aware of.

The $CO_2$ price for Figure 7a is at 18 €t. This information was indeed missing and will be added. Thank you for the notification.

---

## Author Comment (AC2) · 20 May 2020

Thank you for this detailed and very specific review. All points raised are clearly understandable and will be discussed below.

The general scope and purpose of the presented model is to deliver macroscopic long-term trend estimates for a given electricity exchange market at a comparatively slim data demand and low computational cost. Along other (partly strong) simplifications there is no spatial resolution of the generation units. Ultimately, this makes it possible to make a statement on the development of electricity exchange prices without having to solve an optimization problem first. This is of course at the expense of less

detailed model results. Nevertheless, this approach helps to analyze possible future challenges for non-subsidized wind power plants at adequate accuracy as shown with the backtesting in Section 2.6.

1) Expansion of literature review by (equilibrium) energy system models:

As with all commercial tools, PLEXOS has the problem of general accessibility of data and code. Moreover, the possibility of subsequent model adaptation and improvement does not seem to be given. Balmorel was not known to the authors at the time of writing this paper. It appears to be a very interesting tool that will be included in the discussion in the revised version of the paper (Section 1.2). Balmorel has been directed towards the solution of an optimization problem in GAMS. A higher technical level of detail is expected, which in turn leads to increased model complexity and therefore effort compared to the presented model.

2) Clarification of time series data used:

2a) Weather data: Historical weather data for the years 1985 to 2016 are used under the assumption that there will be no significant weather trends in the occurrence of wind and sun by 2040. The time series data provided by Pfenninger and Staffel is used. They use weather data from global reanalysis models and satellite observations (https://www.renewables.ninja/about).

2b) Are any changes in the capacity factors of wind and solar generations considered towards 2040? Not yet. This would be a very interesting extension for a follow-up study. Right now, the energy provided from wind and solar only scales with the overall installed capacity.

2c) How is the load profile modelled and is the same profile used up until 2040? Exactly. The total annual demand of a country is used as the input variable. This parameter is much easier to obtain than an entire time series. Uniform profiles are then used for the hourly variations, which include medium-term (daily and weekly cycles) and longterm (seasonal) effects (see Section 2.2). These profiles are derived from entso-e data (TYNDP18).

2d) Are wind, solar and load profiles synchronized? Wind and solar are synchronized. The same weather year from the studies of Staffel and Pfenninger is used for these technologies. The demand is not synchronized. It was found that especially in Germany the influence beyond the cycles described in 2.2, especially the outdoor temperature, is negligible considering the application area of this model. In countries with high electricity demand for cooling of buildings like Australia this may be different (see Hyndman & Fan, 2010, DOI: 10.1109/TPWRS.2009.2036017)

3-5) How is annual energy consumption and cross-border capacity assumed to change towards 2040?

The simplifications described at the beginning of this response also include an assumption of temporal constancy for most model variables. This includes the total annual electricity demand, commodity prices, export and import (cross-border) capacities as well as generation capacities in neighboring countries. The temporal changes of all these variables are currently already possible and have been carried out in part. In this paper, however, it was intended to first present and discuss the basic function of the model. Therefore, only the installed plant capacities and emission prices were varied. It is undisputed that electricity demand is a very important model parameter whose variation over time should be displayed and investigated. An influence of P2X or electric vehicles on the demand can be made in the assumption of the demand development and is reasonable and intended for future work. For example, a simplified assumption could be made that demand will increase in proportion to the market penetration of electric vehicles. However, a flexibilization of demand would lead to an optimization problem and therefore require a much more extensive adaptation of the model.

6) How are different generation mixtures between countries considered?

Before the main simulation takes place, a pre-simulation is executed for every neighboring country based on the momentary power plant portfolio and total annual demand. This pre-simulation provides the assumed generation mix for each individual neighbouring country and every hour of the forecast period.

7) Can the model capture the likelihood of the highest spot prices?

The absolute price peak at 300 €MWh is in fact hit by the model. So, the curves do not diverge. Rather, the model results show a more regressive course in the area of high market prices. This is probably due to the assumptions made for marginal generation cost of the corresponding conventional power plants (oil and gas).

8) Marketing revenue vs market revenue:

The term is owed to the translation from German into English. In fact, "market revenue" is much more accurate and fitting. Thank you very much for this hint.

9) Addition to figure 8 and PPA prices used:

In contrast to the exchange price of electricity, the remuneration under a PPA is negotiated bilaterally between the contracting parties. However, the negotiation is also based on the exchange price, with the difference being charged to the increased security. The two prices quoted are in fact fictitious. They can be understood to mean that, for example, the WTG operator prepares possible price concepts prior to PPA price negotiations and compares these with various exchange price scenarios for valuation purposes. When using LROE and depending on the assumed PPA mechanism, it is not trivially possible to compare the average stock exchange price with possible PPA remunerations. This should be reflected in the explanations around figure 8. The revised version of this paper will contain appropriate additions.

10) Are transmission bottlenecks inside Germany modelled?

As a spatial resolution is not available, the national transmission network including possible network bottlenecks cannot be mapped. This is a strong simplification of the model.

11) Elaboration on LROE:

During the research for this paper hardly any existing literature on the concept of LROE was found, except for one forum article which will be cited in the revised version of the paper. To differentiate between LCOE and LROE, it is assumed that the economic efficiency of a wind turbine can be assessed based on three essential quantities: Costs, market revenues and electricity yield. LCOE considers the costs and the electricity yield for a specific case (a specific plant). The value can be interpreted as the minimum revenue required for an economical plant operation. LROE on the other hand provides information about the financial revenue potential in a given market as well as for given site conditions. The value is therefore not just a plant information, but it also considers the market in which the plant is operating in. The main advantage and difference to the LCOE concept is therefore, the additional market information. Furthermore, plant costs and associated uncertainties are not included in the measured variable. The latter also leads to a good transferability to different plant concepts. You are right that the denominators of both sizes are the same. Therefore, for an economic operation of WTGs without subsidies on a market, it must apply that LCOE $\leq$ LROE. Just like LCOE, LROE can be used to define and evaluate technical and financial development goals for engineering. Moreover, it can be used by authorities within future subsidization considerations. The subsidies in the form of the tendering procedure follow the LCOE. Accordingly, a funding which considers the LROE for different technologies would be a more holistic approach and a more indirect technology support.

12) Conclusion section:

It is agreed that the conclusion is not given enough attention in the current version of the publication. In the revised version, this section will be supplemented based on the given reviewers' comments and the response above. Especially the results chapter will be extended.

---

## Author Response (AR1)

**Point-by-point response to all referee comments and corresponding changes to the revised manuscript**

Anonymous Referee #1

Thank you for this nice publication. The paper presents a possibility to forecast the development of energy spot prices with respect to expansion of renewable energies. The paper clearly states the benefit of renewables by using an understandable method.

10  RC1: Nonetheless the paper is missing a clear conclusion. At the moment, the conclusion is mixed with the discussion and outlook.

AR1: *As for the conclusion, it is agreed that this part was not given enough attention in the current version of the publication. The revised version contains a separate conclusion in Section 3.3.*

15  RC2: The scenario selection should be more elaborated and discussed. What would be if the expansion of renewables is decreased overtime. As Space and possibilities are limited. What is the likelihood of both scenarios?

AR2: *During this paper it was intended to present and discuss the given model with a first sensitivity study towards emission prices and a brief case study. For this purpose, two simple expansion scenarios have been used where Scenario A only depicts expansion in accordance with the targets of the renewable energy act and therefore is solely theoretical. Scenario B was given*

20  *by entso-e results and might be possible to happen. Anyhow, a broadened scenario analysis with further expansion possibilities would not fit the scope of this paper, but rather have the potential for another interesting investigation and downstream application of the given model. Minor additions have been made to Section 3.1.*

RC3: Section 2.5 should be started with the idea that cross-border interaction are modelled by as power plants. However, what

25  happens if the neighboring countries are demanding instead of delivering?

AR3: *By handling neighboring countries as calculatory power plants within the merit order approach it is indeed possible to display the case that a neighboring country is demanding. This is described in Section 2.5 alongside Figure 3. The revised version of this paper includes further explanations on this subject. The corresponding section is now started as proposed with the idea that cross-border interaction are modelled by as power plants*

30

RC4: In Section 2.6, why did the model not test on a single country? You mentioned the markets have been separated in 2018.

AR4: *During the time of this papers emergence there was simply no data for the sole German market available for an entire year (2019). Application on the separated market is intended for future works. Therefore, no changes have been made for this subject.*

35

RC5: I personally have my doubts that you are the first one to think of LROE. I assume the citation is missing here.

AR5: *Hardly any existing literature on the concept of LROE was found, except for one forum article which is now cited in Section 3. Moreover, the discussion of LROE has been expanded in this section. Please let us know if and where there is scientific literature about this topic that the authors are not aware of.*

40

RC6: Rephrasing of line 220, the content does not come across.

AR6: *The former line 220 (now line 294 and 295) has been rephrased for enhanced comprehensibility.*

RC7: Figure 7a: what is the Co2 price for these graphs?

45   AR7: *The CO2 price for Figure 7a is at 18 €/t. This information was indeed missing and was added to the caption of Figure 7a in line 358.*

RC8: Overall, the paper is missing some comma and might be rephrased at certain points. Examples: - have to = must - Finally, ... - By forecasting future prices, it is ...

50   AR8: *These mistakes have been corrected throughout the document.*

RC9: Table design is not consistent

AR9: *Table design has been adapted.*

55 Matti Koivisto (Referee)

mkoi@dtu.dk

In the paper "Future Economic Perspective and Potential Revenue of Non-Subsidized Wind Turbines in Germany", the authors present a model for estimating future spot market prices in Germany. Two different scenarios towards 2040 are then presented,

60 considering also different CO2 prices. Revenue from a case study wind power plant is compared for different CO2 price scenarios, with a different PPA purchase prices also considered. The paper presents a very important analysis, considering the possible challenges of non-subsidized wind power plants in the future being able to generate sufficient revenue. However, I ask the authors to go through the comments below. The authors should clarify especially the LROE concept, and provide clearer conclusions of the presented analyses (these are the two final comments).

65

RC1: The authors provide a comparison of the presented electricity price model to literature, but focus mainly on regression/time series type modelling. However, there are methods similar to the presented model implemented in energy system modelling tools, such as PLEXOS (https://energyexemplar.com/solutions/plexos/) or Balmorel (http://www.balmorel.com/; code freely available). It would be beneficial if the authors could expand the literature review to

70 one/some of the energy system modelling tools (some open tools: https://en.wikipedia.org/wiki/Open_energy_system_models).

*AR1: As with all commercial tools, PLEXOS has the problem of general accessibility of data and code. Moreover, the possibility of subsequent model adaptation and improvement does not seem to be given. Balmorel was not known to the authors at the time of writing this paper. It appears to be a very interesting tool that is now included in the discussion in Section 1.2.*

75 *Balmorel has been directed towards the solution of an optimization problem in GAMS. A higher technical level of detail is expected, which in turn leads to increased model complexity and therefore effort compared to the presented model.*

RC2: The authors should clarify how the hourly profiles of load and wind and solar generation are modelled towards 2040:

2.a) Wind and solar: In section 3.1., it is said "The annual fluctuations are merely due to different weather conditions in the

80 individual years". But it is not clear to me what weather conditions are assumed, e.g., for 2035 or 2040 (this is not clear from section 2.4). Please clarify (and consider expanding section 2.4).

*AR2a: Historical weather data for the years 1985 to 2016 are used under the assumption that there will be no significant weather trends in the occurrence of wind and sun by 2040. The time series data provided by Pfenninger and Staffel is used. They use weather data from global reanalysis models and satellite observations (https://www.renewables.ninja/about).*

85 *Section 2.4 has been expended accordingly.*

2.b) Are any changes in the capacity factors of wind and solar generations considered towards 2040, as additional installations (presumably with different technologies, e.g., hub height and turbine type) appear to the system?

*AR2b: Not yet. This would be a very interesting extension for a follow-up study. Right now, the energy provided from wind*
90 *and solar only scales with the overall installed capacity. No changes have been made.*

2.c) How is load profile modelled for the different years towards 2040? Is the same profile (i.e., time series shape over the year) assumed all the way to 2040, with only the annual energy level changing? Please clarify.

*AR2c: Exactly. The total annual demand of a country is used as the input variable. This parameter is much easier to obtain*
95 *than an entire time series. Uniform profiles are then used for the hourly variations, which include medium-term (daily and weekly cycles) and long-term (seasonal) effects (see Section 2.2). These profiles are derived from entso-e data (TYNDP18).*

2.d) Are wind, solar and load profiles synchronized? I.e., is for example 2035 defined so that the wind, solar and load data are based on the same weather year (as they all might be correlated due to weather dependencies)?

100 *AR2d: Wind and solar are synchronized. The same weather year from the studies of Staffel and Pfenninger is used for these technologies. The demand is not synchronized. It was found that especially in Germany the influence beyond the cycles described in Section 2.2, especially the outdoor temperature, is negligible considering the application area of this model. In countries with high electricity demand for cooling of buildings like Australia this may be different (see Hyndman & Fan, 2010, DOI: 10.1109/TPWRS.2009.2036017). Section 2.4 has been adapted for enhanced comprehensibility.*

105

RC3: How is annual energy consumption assumed to change towards 2040? As this is a quite fundamental variable for the system, its progression towards 2040 would be nice to see as a figure or table.

*AR3: The general scope and purpose of the presented model is to deliver macroscopic long-term trend estimates for a given electricity exchange market at a comparatively slim data demand and low computational cost. Along other simplifications*
110 *there is no spatial resolution of the generation units. Also, many dynamic parameters (including the total annual electricity demand) are assumed to be static. Ultimately, this makes it possible to make a statement on the development of electricity exchange prices without having to solve an optimization problem first. This is of course at the expense of less detailed model results. The assumption regarding total annual electricity demand has been added to Section 2.2.*

115 RC4: Considering consumption changes, are, e.g., power-to-gas, electrification of heating and/or electric vehicles considering going towards 2040?

*AR4: It is undisputed that electricity demand is a very important model parameter whose variation over time should be displayed and investigated in future studies. An influence of P2X or electric vehicles on the demand can be made in the assumption of the demand development and is reasonable and intended for future work. For example, a simplified assumption*
120 *could be made that demand will increase in proportion to the market penetration of electric vehicles. However, a flexibilization*

*of demand would lead to an optimization problem and therefore require a much more extensive adaptation of the model. As the discussed extensions would go beyond the scope of this paper, no changes have been made in the revised document.*

RC5: Are cross-border capacities (Table 3) assumed to remain on the 2018 level all the way to 2040? Is this justifiable?

125 *AR5: Next to annual electricity demand, the assumption of static parameters mentioned in AR3 is also applied to commodity prices, export, and import (cross-border) capacities as well as generation capacities in neighboring countries. The temporal changes of all these variables are currently already technically possible and have been carried out in part. In this paper, however, it was intended to first present and discuss the basic function of the model. Therefore, no changes have been made in the revised document. Since cross-border capacities will most definitely be expanded in reality, this would be another*

130 *interesting follow-up investigation.*

RC6: In section 2.5, it reads "To estimate power import and export every neighboring country is modelled as a hypothetic power plant with individual capacity and marginal cost". Please elaborate a little more on this. E.g., how are the very different generation mixtures in the different countries taken into account (hydro in NO, nuclear in FR, and so on)?

135 *AR6: Before the main simulation takes place, a pre-simulation is executed for every neighboring country based on the momentary power plant portfolio and total annual demand. This pre-simulation provides the assumed generation mix for each individual neighbouring country and every hour of the forecast period. Section 2.5 was extended accordingly.*

140 RC7: About Figure 4: It seems that the model cannot capture the likelihoods of the highest spot prices (i.e., the lines diverge going to the left). Can any discussion be given on to why this happens?

*AR7: The absolute price peak at 300 €/MWh is in fact hit by the model. So, the curves do not diverge. Rather, the model results show a more regressive course in the area of high market prices. This is probably due to the assumptions of marginal generation costs of the corresponding conventional power plants (oil and gas).*

145

RC8: Word "marketing revenue" is used in many places. I find it a little bit confusing, as "marketing" usually refers to advertisement. Perhaps "market revenue" could be used? (please disregard this comment if "marketing revenue" is an often used term in this context)

*AR8: The term was owed to the translation from German into English. In fact, "market revenue" is much more accurate and*

150 *has been replaced throughout the revised document.*

RC9: Understanding Figure 8: The different CO2 prices for the orange bars provide a clear comparison. However, it is not clear to me how the different PPA prices should be considered. How are the 40 CMWh and 50 CMWh linked to the studied

expansion scenario B and the different CO2 prices? In the current form, it seems that the 40 CMWh and 50 CMWh are arbitrary prices, and therefore comparisons between direct market revenues and PPAs is difficult.

*AR9: In contrast to the exchange price of electricity, the remuneration under a PPA is negotiated bilaterally between the contracting parties. However, the negotiation is also based on the exchange price, with the difference being charged to the increased security. The two prices quoted are in fact fictitious. They can be understood to mean that, for example, the WTG operator prepares possible price concepts prior to PPA price negotiations and compares these with various exchange price scenarios for valuation purposes. When using LROE and depending on the assumed PPA mechanism, it is not trivially possible to compare the average stock exchange price with possible PPA remunerations. This should be reflected in the explanations around figure 8. The revised version of this paper's Section 3.2 contains the corresponding extensions.*

RC10: Are transmission bottlenecks inside Germany and resulting possible curtailment of wind generation considered/modelled?

*AR10: As a spatial resolution is not available (see assumptions mentioned in AR3), the national transmission network including possible network bottlenecks cannot be mapped. This is a strong simplification.*

RC11: Please elaborate more on the LROE concept. Is it based on existing literature? For LCOE, the top part (dividend) are costs, whereas in LROE they are revenue (the divisor seems to be the same in both, namely energy produced). This seems to be a very significant difference (cost = expense vs. revenue = income). Please discuss a little more on LCOE versus LROE, and why LROE is considered a good measure in the paper. For LCOE, the resulting CMWh value can be understood as the minimum (constant) electricity price over the lifetime to make the project profitable. How resulting LROE values should be understood? (please link the discussion also to the LROE values reported in Figure 8).

*AR11: During the research for this paper hardly any existing literature on the concept of LROE was found, except for one forum article which will be cited in the revised version of the paper. To differentiate between LCOE and LROE, it is assumed that the economic efficiency of a wind turbine can be assessed based on three essential quantities: Costs, market revenues and electricity yield. LCOE considers the costs and the electricity yield for a specific case (a specific plant). The value can be interpreted as the minimum revenue required for an economical plant operation.*

*LROE on the other hand provides information about the financial revenue potential in a given market as well as for given site conditions. The value is therefore not just a plant information, but it also considers the market in which the plant is operating in. The main advantage and difference to the LCOE concept is the additional market information. Furthermore, plant costs and associated uncertainties are not included in the measured variable. The latter also leads to a good transferability to different plant concepts. You are right that the denominators of both sizes are the same. Therefore, for an economic operation of WTs without subsidies on a market, it must apply that LCOE ≤ LROE.*

*Just like LCOE, LROE can be used to define and evaluate technical and financial development goals for engineering. Moreover, it can be used by authorities within future subsidization considerations. The subsidies in the form of the tendering*

*procedure follow the LCOE. Accordingly, a funding which considers the LROE for different technologies would be a more holistic approach and a more indirect technology support.*

190    *The discussion above has been added to the revised document's Sections 3 and 3.2.*

RC12: The Introduction says "A scenario analysis highlights that most of today's wind turbines are not able to yield financial profit over their lifetime without guaranteed subsidies in Germany". However, I don't see this result clearly presented later in the paper. Please provide a clear conclusion section, where each result is presented and explained based on the presented

195    analyses.

*AR12: As for the conclusion, it is agreed that this part was not given enough attention in the current version of the publication. The revised version contains a separate conclusion in Section 3.3.*

[revised manuscript text omitted]

---

## Editor Decision (ED1)

Overall comments
- The core contribution of the paper – a preliminary investigation into how LROE for a wind farm can be calculated to aid in design, PPA calculations, etc – is very interesting. However, weaknesses in how the method is presented, lack of justification for key analysis decisions, incomplete reporting of the results, and inadequate grounding of the work in prior art mean that major revision is still needed.
- The paper structure needs work
    - It needs significant improvement in this work builds from and extends the state of the art – i.e. the contribution beyond the past efforts.
    - The methods section jumps right into details without providing a good overview of the approach and most importantly WHY such an approach has been taken and how it differs and improves upon past efforts
    - The results section provides one results of one type for scenario A and another type for scenario B. This mix and matching tells an incomplete story. Both scenario results need reporting in order to properly build to the conclusions
    - Outlook is insufficient in addressing key paper limitations
- The overall paper needs to be edited for grammar before final publication. At many points it is difficult to interpret the author's intent without re-reading sentences several times.
    - I could not even get through the second page without spotting a large number of syntax and grammar errors. I will not proceed with a thorough review of grammar but will check it prior to final publication

Abstract
- Line 7 - Consecutively incorrect English grammar – perhaps considerably? Or in the near future?
- Line 8 grammar error
- Dynamic field-in profile is odd language – you mean the hourly generation profile?
- Are the dismantling of the plants and emissions prices sensitivities you are looking at in the study? Awkward wording
- Remove last sentence of the abstract – it's an overstatement of the contribution and borderline grandiose. The sentence before is fine and speaks more directly to the potential value of the work

Introduction
- Need to qualify that you are talking about Germany rather than referring only to the title of the act. I recommend rephrasing to something like "Renewable electricity generation has increased exponentially in Germany over the last few decades due in large part to the Renewable Energy Act of (year)…"
- Grammar error in sentence 31-32
    - I will not correct grammar errors beyond this point because there are too many
- Line 42, what state of the art? Lacking a bit in terms of citations… please do not speak generically about other work. Cite a specific work or collection of works

- Line 56 – direct marketing doesn't make sense in English. Since it is key terminology, it is important to update it. Try direct merchant market participation or direct marketing of their electricity to the system, etc…
- PPAs have been the historic status quo in the US for decades. (whereas most of Europe tended towards FITs). There is some missing context here. PPAs are not novel by any means. There are tons of works out there that compare and contrast FITs, PPAs, quotas and other policy support mechanisms. It might be good to provide a bit more of that broader context before jumping to the current debate around PPAs in Germany
- Pg 115 many proprietary models are commercial. I would not call this a criticism. It's a common feature of commercial software. I also agree with the reviewer that even though PLEXOS is a commercial code, it is used extensively both by industry and the research community and should be mentioned.
- In addition, it would also be good to mention literature by Hirth and others looking at the value of wind energy. You might check the reference list of this recent paper: https://www.sciencedirect.com/science/article/abs/pii/S0960148120301531

Methodology and forecasting model
- before diving into the model, the paper is lacking an overview that describes the key structure of the model, highly level i/o, methodology used and perhaps most importantly, the assumptions and limitations of which there are many (as noted by the reviewers) and this should be front and center so the reader has a good idea of what the model is about before diving into the details.
- Also, how does your modellilng approach compare and contrast to the state of the art?
- What do you mean the model designs adds a more agent-based approach? Are you using an agent-based model? Be careful on terminology
- The assumption that the weather data and load profile do not need to be synchronized is inadequately motivated. I am not convinced per the reference to the one study. I have seen even phase shifting by a few hours can result in very different correlation statistics… for this study, you may have been limited, but again, the assumptions are what make good fodder for future work. You should be realistic about the limitations of the current work and very explicit wherever possible
- Model validation is also inadequate. What statistics can you report? How much do they different in time versus the cumulative effect that is seen in the price duration curve? You must have statistics on the errors overall between the simulated and historical time-series

Model application, results and case study
- Again, see prior recommended paper https://www.sciencedirect.com/science/article/abs/pii/S0960148120301531 . LROE is just one potential metric so it should be discussed in comparison to others… for example, sLCOE is too cumbersome for the current approach, etc. It is a good choice, but it needs further context.
- Line 251 – minimium FIXED or AVERAGE revenue…
- Where does scenario B come from? Did you make it up? If so, how did you choose which plants to dismantle? I agree scenario A is not realistic but it leads me to

believe both scenarios are somewhat ad hoc in their creation. Please provide a bit more justification for the development of the scenarios

- Figure 5 graphic quality needs to be improved
- There are a huge number of works creating future energy scenarios. Even if this work is too far along to use these, it would be good to refer to these and again explain better the choices made in this study. DNV GL, BP, IRENA, IEA etc… there are tons of organizations out there looking at future energy mix. See IEA Wind Task 25 for a relatively good source of studies, also see ESIG and their work.
- Variation in CO2 is a sensitivity analysis on carbon price. Would be good to describe it as such. However, there is a problem here as the price of CO2 will have an endogeneous effect on the long term electricity generation mix. ReEDS and other models take this into account. It is important to note this limitation. See works by Trieu Mai and others from NREL with ReEDS or again, see ther various many works by the collective research community of IEA Wind Task 25
- Explain better what you are doing in lines 299-301… fictitious??? Again, decisions in analysis need grounded explanation
- The PPA approach needs much stronger justification especially if you are going to use it in generalized conclusions based on the results as in lines 307-310
- Why do you not have a similar figure 8 for scenario A? or put both scenarios side by side on the same plot?
- And why is figure 6 only for scenario A? this is very strange. You don't need to use histograms. You could use lines and plot both scenarios on the same plots. Or alternative use two sets of plots – one for each scenario

Conclusions
- What does low data requirements and low computational cost mean? Be specific and compare to alternative approaches
  - Also, given the myriad of assumptions made I do not think you can claim this at present. Much more work is needed to establish external validity of this claim
- For conclusions on the effects of renewables, you need both scenarios to be shown and compared and contrasted (see comments above).

Discussion and outlook
- WIFO needs to be defined (no acronyms should be used).
- Again, strike the last sentence – as with the abstract, it is overly broad
- Outlook insufficiently critical of the current limitations… the outlook section is where you should circle back to what the key limitations of the current approach are. This needs to be done at the beginning of the methodology section and then the different approaches to remedy them should be discussed here.

---

## Author Response (AR2)

**Point by point response:**

Associate Editor (E)

Authors (A)

**Overall comments**

**E:** *The core contribution of the paper – a preliminary investigation into how LROE for a wind farm can be calculated to aid in design, PPA calculations, etc – is very interesting. However, weaknesses in how the method is presented, lack of justification for key analysis decisions, incomplete reporting of the results, and inadequate grounding of the work in prior art mean that major revision is still needed.*

**A:** Thank you very much for this very detailed and constructive review. Your comments below are very specific and have highlighted main weaknesses of the paper that needed major improvement. We have tried to implement all comments in the revised version of the paper.

**E:** ***The paper structure needs work.*** *It needs significant improvement in this work builds from and extends the state of the art – i.e. the contribution beyond the past efforts. The methods section jumps right into details without providing a good overview of the approach and most importantly WHY such an approach has been taken and how it differs and improves upon past efforts*

**A:** The derivation of the model requirements from the state of the art and a broad overview of the selected procedure is now presented at the beginning of Section 2. Originally, Figure 1 was intended to fulfill this purpose. This Figure has been revised and now also contains the connection between the results of the forecast model and the LROE.

**E:** *The results section provides one results of one type for scenario A and another type for scenario B. This mix and matching tells an incomplete story. Both scenario results need reporting in order to properly build to the conclusions*

**A:** The results of the respective other scenario are now also listed and analyzed in Section 3. The idea behind the previous version was to present only the most interesting scenario results. However, it is agreed that this has resulted in an incomplete presentation.

**E:** *Outlook is insufficient in addressing key paper limitations*

**A:** The overall presentation of the model has been revised throughout the paper to take a more critical look at its limitations and simplifications. The limitations of the presented method are now explicitly mentioned again in the outlook section.

**E:** *The overall paper needs to be edited for grammar before final publication. At many points it is difficult to interpret the author's intent without re-reading sentences several times. I could not even get through the second page without spotting a large number of syntax and grammar errors. I will not proceed with a thorough review of grammar but will check it prior to final publication*

**A:** The grammar in the paper was revised. Please excuse the previous mistakes.

**Abstract**

**E:** *Line 7 - Consecutively incorrect English grammar – perhaps considerably? Or in the near future?*

**E:** *Line 8 grammar error*

**A:** Again, please apologize for the inconvenience. Most grammar mistakes have been due to translation and working in the revision mode of MS Word. The grammar has been revised.

**E:** *Dynamic field-in profile is odd language – you mean the hourly generation profile?*

**A:** Because of the slight difference between "generated electricity" and "electricity fed into the grid", the term "feed-in" was originally used throughout the paper. In retrospect, it is agreed that this wording is somewhat strange in English. It has been corrected in the revised document towards "electricity generation".

**E:** *Are the dismantling of the plants and emissions prices sensitivities you are looking at in the study? Awkward wording*

**A:** Yes. The wording in the abstract was adapted to make it more understandable.

**E:** *Remove last sentence of the abstract – it's an overstatement of the contribution and borderline grandiose. The sentence before is fine and speaks more directly to the potential value of the work*

**A:** The last sentence has been removed. The tone of the entire paper has been adjusted towards a more neutral and less glorifying language to present the overall model results.

**Introduction**

**E:** *Need to qualify that you are talking about Germany rather than referring only to the title of the act. I recommend rephrasing to something like "Renewable electricity generation has increased exponentially in Germany over the last few decades due in large part to the Renewable Energy Act of (year)…"*

**A:** The beginning of the introduction has been adapted accordingly.

**E:** *Grammar error in sentence 31-32*

**A:** The introduction has been revised in terms of correct grammar.

**E:** *Line 42, what state of the art? Lacking a bit in terms of citations… please do not speak generically about other work. Cite a specific work or collection of works*

**A:** This sentence has been moved to the discussion section and refers now to the state-of-the-art models discussed and cited in Section 1.2.

**E:** *Line 56 – direct marketing doesn't make sense in English. Since it is key terminology, it is important to update it. Try direct merchant market participation or direct marketing of their electricity to the system, etc…*

**A:** Like for the term "feed-in" the terminology "direct marketing" has been replaced by "selling electricity directly on the exchange markets" in the entire manuscript to enhance comprehensibility.

**E:** *PPAs have been the historic status quo in the US for decades. (whereas most of Europe tended towards FITs). There is some missing context here. PPAs are not novel by any means. There are tons of works out there that compare and contrast FITs, PPAs, quotas and other policy support mechanisms. It might be good to provide a bit more of that broader context before jumping to the current debate around PPAs in Germany*

**A:** The globally different propagation and prior application of PPAs is now described in more detail in Section 1.1 to provide more context.

**E:** *Pg 115 many proprietary models are commercial. I would not call this a criticism. It's a common feature of commercial software. I also agree with the reviewer that even though PLEXOS is a commercial code, it is used extensively both by industry and the research community and should be mentioned.*

**A:** Being commercial is less of a critical point rather than not knowing, how calculations are being executed or what data is used. Because PLEXOS seems to be very transparent in terms of underlying data sets and calculation methods and because of its broad applications it is now being mentioned in the state of the art in Section 1.2. It is also used for benchmark in terms of calculation time in Section 2.6.

**E:** *In addition, it would also be good to mention literature by Hirth and others looking at the value of wind energy. You might check the reference list of this recent paper: https://www.sciencedirect.com/science/article/abs/pii/S0960148120301531*

**A:** A new Section 1.3 has been added, covering some relevant measurands for assessing the economic efficiency and value of wind energy, also including Hirth's work and the recommended recent paper. The reference list was found to be particularly helpful in finding some more very interesting documents for this research.

**Methodology and forecasting model**

**E:** *before diving into the model, the paper is lacking an overview that describes the key structure of the model, highly level i/o, methodology used and perhaps most importantly, the assumptions and limitations of which there are many (as noted by the reviewers) and this should be front and center so the reader has a good idea of what the model is about before diving into the details.*

**A:** Section 2.1 now contains a high-level method description and the most essential model assumptions. Detailed assumptions are presented in the respective sub-sections together with the corresponding modelling parameters.

**E:** *Also, how does your modellilng approach compare and contrast to the state of the art?*

**A:** The modelling approach is now put into perspective by model classifications by Weron* and compared to the models cited in Section 1.2. However, in accordance with Weron*, the comparison of different models' results was found to be difficult due to deviations in the forecast object and granularity of the model structure and calculation method. This is why the backtesting approach has been chosen for validation. A short comparison to PLEXOS in terms of calculation time has been added in Section 2.6. Anyhow, further comparison should be done in future model application. This is now also mentioned in the outlook.
*https://www.researchgate.net/publication/265853980_Electricity_price_forecasting_A_review_of_the_state-of-the-art_with_a_look_into_the_future

**E:** *What do you mean the model designs adds a more agent-based approach? Are you using an agent-based model? Be careful on terminology*

**A:** A single object with an individual cost function is derived for each conventional power plant, which can also be individually parameterized and analyzed later. The number of power plants constitutes the number of agents. Because of the underlying a merit-order approach, the decision-making heuristic for every agent and hour is to offer electricity at their own marginal generation cost and to generate and sell electricity if they are below the uniform market clearing price. An interaction topology is given by the competition to the other power plants or agents. The exogenously given electricity demand represents the environmental influence that drives the

decision of every agent. Even though this poses a very simple multi-agent approach, it can be classified as such. For future studies, learning rules for agents as well as the introduction of randomness could be very interesting. This argumentation has been added to section 2.3 and the outlook of the manuscript.

**E:** *The assumption that the weather data and load profile do not need to be synchronized is inadequately motivated. I am not convinced per the reference to the one study. I have seen even phase shifting by a few hours can result in very different correlation statistics... for this study, you may have been limited, but again, the assumptions are what make good fodder for future work. You should be realistic about the limitations of the current work and very explicit wherever possible.*

**A:** The investigation regarding correlation between temperature and demand has been added to Section 2.4. The limitations that come with the simplification of desynchronized weather data and demand are now part of the discussion section.

**E:** *Model validation is also inadequate. What statistics can you report? How much do they different in time versus the cumulative effect that is seen in the price duration curve? You must have statistics on the errors overall between the simulated and historical time-series*

**A:** Designed for a long-term forecast, the model is not suitable for estimating exact hourly price patterns. This is partly due to the neglection of weather effects and the annually recurring fluctuation in demand. Anyhow, overall price trends and distributions can be reconstructed, which was the initial requirement for the research question. Statistics for both comparisons are now stated and discussed in Section 2.6.

**Model application, results and case study**

**E:** *Again, see prior recommended paper https://www.sciencedirect.com/science/article/abs/pii/S0960148120301531 . LROE is just one potential metric so it should be discussed in comparison to others... for example, sLCOE is too cumbersome for the current approach, etc. It is a good choice, but it needs further context.*

**A:** Further metrics are now described and discussed in the new Section 1.3. The decision for LROE is being argued in the revised Section 2.

**E:** *Line 251 – minimium FIXED or AVERAGE revenue...*

**A:** Fixed. The addition has been made in the revised version and discussion of LCOE in Section 1.3.

**E:** *Where does scenario B come from? Did you make it up? If so, how did you choose which plants to dismantle? I agree scenario A is not realistic but it leads me to believe both scenarios are somewhat ad hoc in their creation. Please provide a bit more justification for the development of the scenarios*

**A:** Additional information on how both scenarios have been developed is now provided and cited in Section 3. The scenarios are based on the renewable expansion path as defined by the Renewable Energy Act and on the scenario framework approved by the German Federal Network Agency in June 2018.

**E:** *Figure 5 graphic quality needs to be improved*

**A:** The graphic has been replaced at a better resolution.

**E:** *There are a huge number of works creating future energy scenarios. Even if this work is too far along to use these, it would be good to refer to these and again explain better the choices made in this study. DNV GL, BP, IRENA, IEA etc… there are tons of organizations out there looking at future energy mix. See IEA Wind Task 25 for a relatively good source of studies, also see ESIG and their work.*

**A:** Further scenarios developed by the different organizations are now referred to in Section 3 and in the outlook of the revised paper. Thanks again for pointing at the relevant sources.

**E:** *Variation in CO2 is a sensitivity analysis on carbon price. Would be good to describe it as such. However, there is a problem here as the price of CO2 will have an endogeneous effect on the long term electricity generation mix. ReEDS and other models take this into account. It is important to note this limitation. See works by Trieu Mai and others from NREL with ReEDS or again, see ther various many works by the collective research community of IEA Wind Task 25*

**A:** The wording for the sensitivity analysis has been adapted. Interdependence between emission price and generation mix can not be considered since the generation capacities need to be pre-defined for each scenario. This, among the other limitations, is now part of the model discussion.

**E:** *Explain better what you are doing in lines 299-301… fictitious??? Again, decisions in analysis need grounded explanation*

**A:** Analysis decision and presentation have been revised for Section 3.2. This line was only hypothetical and fictious and has therefore been removed for better comprehensibility.

**E:** *The PPA approach needs much stronger justification especially if you are going to use it in generalized conclusions based on the results as in lines 307-310*

**A:** Further PPA considerations have been added to Section 1.1. The formulations in the revised paper have been weakened in a less generalizing way.

**E:** *Why do you not have a similar figure 8 for scenario A? or put both scenarios side by side on the same plot?*

**A:** A new Figure 8(a) has been added that addresses the case study results for scenario A.

**E:** *And why is figure 6 only for scenario A? this is very strange. You don't need to use histograms. You could use lines and plot both scenarios on the same plots. Or alternative use two sets of plots – one for each scenario*

**A:** Figure 6 has been revised and shows now the results for both scenarios.

**Conclusions**

**E:** *What does low data requirements and low computational cost mean? Be specific and compare to alternative approaches. Also, given the myriad of assumptions made I do not think you can claim this at present. Much more work is needed to establish external validity of this claim*

**A:** Currently, a complete model execution takes about five minutes. Regarding the data requirements, only the annual cumulated installed generation capacity of the power plants and the annual electricity demand for the forecast period are needed for the current calculations of different scenarios. All other dynamic quantities are determined automatically or are obtained from the presented reference data of the TYNDP. Nevertheless, you are right, that because of the many assumptions this claim needed to be relativized and revised.

**E:** *For conclusions on the effects of renewables, you need both scenarios to be shown and compared and contrasted (see comments above).*

**A:** The model results have been added and discussed accordingly in Section 3.1.

**Discussion and outlook**

**E:** *WIFO needs to be defined (no acronyms should be used).*

**A:** WIFO is defined now in the outlook. It stands for WInd Farm Optimizer.

**E:** *Again, strike the last sentence – as with the abstract, it is overly broad*

**A:** The sentence has been removed.

**E:** *Outlook insufficiently critical of the current limitations... the outlook section is where you should circle back to what the key limitations of the current approach are. This needs to be done at the beginning of the methodology section and then the different approaches to remedy them should be discussed here.*

**A:** The discussion and outlook section has been revised more critically, especially in terms of the current model assumptions and limitations as well as necessary model improvements to dissolve simplifications.

**Overview of major changes:**

- The Paper has been revised for grammar errors and overly exalting language.
- The Introduction (Section 1) has been revised to better derive the papers research question and subsequent model.
- PPAs are now given a more global discussion in Section 1.1 before mentioning the current discussion in Germany.
- A new Section 1.2 has been added, addressing different metrices for evaluating profitability of renewable energy sources.
- The model PLEXOS has been integrated in the regarded state of the art models and subsequent discussion.
- Section 2 has been revised and now contains a broader model description including the most relevant assumptions.
- The argumentation about desynchronized weather data and demand has been extended in Section 2.5.
- Model validation has been revised by additionally discussing further error quantities as well as the temporal course of the prices.
- The chosen scenarios in Section 3 are now derived more thoroughly. Results are now presented and discussed for both scenarios throughout all Sub-Sections.
- The discussion in Section 4 now deals much more critically with model limitations and assumptions as well as future improvement measures.

**Marked changes:**

[revised manuscript text omitted]